# A globally calibrated scheme for generating daily meteorology from monthly statistics: Global-WGEN (GWGEN) v1.0

Philipp S. Sommer[1] and Jed O. Kaplan[1,2]

[1]Institute of Earth Surface Dynamics, University of Lausanne, Géopolis, 1015 Lausanne, Switzerland
[2]Max Planck Institute for the Science of Human History, 07745 Jena, Germany

*Correspondence to:* Philipp S. Sommer (philipp.sommer@unil.ch)

**Abstract.** While a wide range of earth system processes occur at daily and even sub-daily timescales, many global vegetation and other terrestrial dynamics models historically used monthly meteorological forcing, both to reduce computational demand and because global datasets were lacking. Recently, dynamic land surface modeling has moved towards resolving daily and subdaily processes, and global datasets containing daily and sub-daily meteorology have become available. These meteorological datasets, however, cover only the instrumental era of the last ca. 120 years at best, are subject to considerable uncertainty, and represent extremely large data files with associated computational costs of data input/output and file transfer. For periods before the recent past or into the future, global meteorological forcing can be provided by climate model output, but the quality of these data at high temporal resolution is low, particularly for daily precipitation frequency and amount. Here we present GWGEN, a globally applicable statistical weather generator for the temporal downscaling of monthly climatology to daily meteorology. Our weather generator is parameterized using a global meteorological database and simulates daily values of five common variables: minimum and maximum temperature, precipitation, cloud cover, and wind speed. GWGEN is lightweight, modular, and requires a minimal set of monthly mean variables as input. The weather generator may be used in a range of applications, for example, in global vegetation, crop, soil erosion, or hydrological models. While GWGEN does not currently perform spatially autocorrelated multi-point downscaling of daily weather, this additional functionality could be implemented in future versions.

## 1 Introduction

The development of the first global vegetation models in the 1970's (e.g., Lieth, 1975) brought about the demand for meteorological forcing datasets with global extent and relatively high spatial resolution, e.g., $1° \times 1°$. While a global weather station-based monthly climate dataset was available at this time (Walter and Lieth, 1967), limitations in computers and storage allowed only the simplest treatment of these data. The first global simulations of the net primary productivity of the terrestrial biosphere (Lieth, 1975), thus used rasterized polygons of annual meteorological variables that had been crudely interpolated from the station-based climatology. A decade later saw the development of better computers and more sophisticated global vegetation models (Prentice et al., 1992; Prentice, 1989) that recognized the need for forcing at a sub-annual timestep and development of these models was done in parallel with the first global, gridded high resolution (0.5°) monthly climatology

(Leemans and Cramer, 1991). At the time, monthly meteorological data was the only feasible global data that could be produced, in terms of the raw station data available to feed the interpolation process, the processing time required to produce gridded maps, and the data storage and transfer capabilities of contemporary computer systems and networks. Global gridded monthly climate data thus became the standard for not only large-extent vegetation modeling (Haxeltine and Prentice, 1996; Haxeltine et al., 1996; Kaplan et al., 2003; Kucharik et al., 2000; Woodward et al., 1995), but also for a wide range of studies on biodiversity and species distribution (e.g., Elith et al., 2006), vegetation trace gas emissions (e.g., Guenther et al., 1995), and even the geographic distribution of human diseases (e.g., Bhatt et al., 2013)

Over subsequent years, the global gridded monthly climate datasets were improved (New et al., 1999, 2002), developed with very high spatial resolution (Hijmans et al., 2005), and expanded beyond climatological mean climate to cover continuous timeseries over decades (Harris et al., 2014; Mitchell and Jones, 2005; New et al., 2000). The latter was an essential requirement for forcing dynamic global vegetation models (DGVMs) (e.g., Sitch et al., 2003). However, despite increasing quality, spatial resolution, and temporal extent in these datasets, the basic time step remained monthly, partly for legacy reasons — models had been developed in an earlier era subject to computational limitations and therefore used a monthly timestep for efficiency even if this was no longer strictly a constraint — and partly because of the challenge in developing a global, high-resolution climate dataset with a daily or shorter timestep still presented a major data management challenge.

On the other hand, there was increasing awareness that accurate simulation of many earth surface processes required representation of processes at a shorter-than-monthly timestep. Global simulation of surface hydrology (Gerten et al., 2004), crop growth (Bondeau et al., 2007), or biogeophysical processes (Krinner et al., 2005) needed sub-monthly forcing to produce reliable results. To address this need for better forcing data, two main approaches were taken: either monthly climate data were downscaled online using a stochastic weather generator (e.g., Pfeiffer et al., 2013), or a sub-daily, high-resolution, gridded climate timeseries was generated directly by merging high-temporal-resolution reanalysis data (e.g., NCEP, 6h, 2.5°) with high-spatial-resolution monthly climate data (e.g., CRU, 0.5°). The latter process resulted in the CRUNCEP dataset (Viovy and Ciais, 2016; Wei et al., 2014), which, while global, is large even by modern standards (ca. 350 Gb), is not available at spatial resolution greater than 0.5°, and covers only the period 1901-2014.

Forcing data for global vegetation and other models with shorter than monthly resolution at higher spatial resolutions than 0.5°, or for any other period than the last ca. 120 years, e.g., for the future or the more distant past, may therefore only be available through downscaling techniques. One approach to overcome the limitations of currently available datasets could be to use GCM output directly, however, most GCM output currently available does not have greater than 0.5° spatial resolution, with the current generation of GCMs typically approaching ca. $1° \times 1°$. Furthermore, there is a general observation that daily meteorology produced by GCMs is not realistic, particularly for precipitation (Dai, 2006; Stephens et al., 2010; Sun et al., 2006). An alternative approach is, therefore, to perform temporal downscaling on monthly meteorological data using a statistical weather generator.

Statistical weather generators were first developed primarily for crop and hydrological modeling at the field to catchment scale (Richardson, 1981; Woolhiser and Pegram, 1979; Woolhiser and Roldan, 1982). The weather generator was parameterized using daily meteorological observations at one or more weather stations close to the area of interest, although some attempts

were made to generalize the parameterization over larger, sub-continental regions (e.g., Wilks, 1999b, 1998; Woolhiser and Roldán, 1986). Locally parameterized weather generators have been applied to a very wide range of studies (Wilks, 2010; Wilks and Wilby, 1999), and enhanced to include additional meteorological variables beyond the original precipitation, temperature, and solar radiation (e.g., Parlange and Katz, 2000). Applications of a weather generator at continental to global scales was still limited, however, because of the need to perform local parameterization.

The need to simulate daily meteorology in regions of the world with short, unreliable, or unavailable daily meteorological timeseries brought about the realization that certain features of weather generator parameterization might be generalized across a range of climates (Geng and Auburn, 1987; Geng et al., 1986). This ultimately led to the development of globally applicable weather generators (Friend, 1998), and their incorporation in DGVMs (Bondeau et al., 2007; Gerten et al., 2004; Pfeiffer et al., 2013). The original global parameterization (Geng et al., 1986) of these weather generators was, however, limited to seven weather stations, mostly in the temperate latitudes. Friend (1998) does not publish the parameters used in his global weather generator, but we assume these were the same as the original Geng and Auburn (1987) and Geng et al. (1986) models. Given the availability of 1) large datasets of daily meteorology, and 2) computers powerful enough to process these data, we therefore decided that it would be valuable to revisit these parameterizations, perform a systematic and quantitative evaluation of the resulting downscaled meteorology, and potentially improve our ability to perform monthly-to-daily downscaling of common meteorological variables with a single, globally applicable parameterization.

In the following sections we describe Global-WGEN (GWGEN), a weather generator parameterized using more than 50 million daily weather observations from all continents and latitudes. We demonstrate how updated schemes for simulating precipitation occurrence and amount, and for bias correcting wind speed, further improve the quality of the model simulations. We perform an extensive model evaluation and parameter uncertainty analysis in order to settle on a parameter set that provides the most accurate, globally applicable results. We comment on the limitations of the model and priorities for future research. GWGEN is an open-source, stand-alone model that may be incorporated into any number of models designed to work at global scale, including, e.g., vegetation, hydrology, climatology, and animal distribution models.

## 2 Model description

GWGEN requires the following six monthly summary values as input: 1) total monthly precipitation, 2) the number of days in the month with measurable precipitation (i.e., wet days), 3-4) monthly mean daily minimum and maximum temperature, 5) mean cloud fraction, and 6) wind speed. The model outputs are the same variables at daily resolution. This section summarizes the basic workflow in the model which is also shown schematically in Figure 1 and Algorithm 1.

The first approximation of the daily variables comes from smoothing the monthly time series using a mean-preserving algorithm (Rymes and Myers, 2001).

For precipitation we then first use the Markov Chain approach (section 3.2.1) to decide the wet/dry state of the day. If it is a wet day, we calculate the gamma parameters using the equations (7) and (8). The resulting distribution allows us to draw a

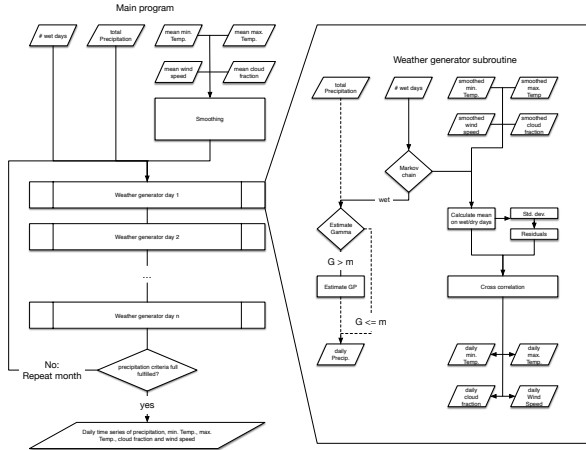

**Figure 1.** Schematic workflow of GWGEN. After smoothing the monthly input, the Markov Chain is used to decide, whether it is a dry or a wet day. If it is a wet day, we draw a random number from the Gamma-GP distribution. Furthermore, the other means of the variables $(\bar{T}_{\min/\max}, \bar{c}, \bar{w})$ are adjusted and their daily values are calculated using the estimated standard deviations and residuals. The wind speed furthermore undergoes a square root transformation before applying the cross correlation and in the end is corrected using the bias correction. A quality check in the end restricts our model to be within a $5\%$ range of the observed total precipitation and to replicate the number of wet days from the input.

random number, the precipitation amount of the currently simulated day. If we are above the threshold $\mu$, we draw a second random number from the GP distribution parameterized via equation (9) and the chosen GP shape.

The next step modifies the means of temperature, wind speed and cloud fraction depending on the wet/dry state of the day (lines 11 and 15 in algorithm 1). After that, we use the cross-correlation approach described in Richardson (1981) (lines 18 -
20 and subsubsection 3.2.6) and calculate the daily values of these variables. Finally we use the quantile-based bias correction described in section 3.4 to correct the simulated wind speed.

We restrict the weather generator to reproduce the exact number of wet days ($\pm 1$) as the input and to be within a $5\%$ range of the total monthly precipitation (with a maximum allowed deviation of $0.5\,\mathrm{mm}$). If the program cannot produce these results, the procedure described above is repeated (see line 4).

**3   Model development**

GWGEN is based on the WGEN weather generator (Richardson, 1981), using the method of defining the model parameters based on monthly summaries described by Geng et al. (1986) and Geng and Auburn (1987). GWGEN diverges from the original WGEN by using a hybrid-order Markov chain to simulate precipitation occurrence (Wilks, 1999a), and a hybrid Gamma-GP distribution (Furrer and Katz, 2008; Neykov et al., 2014) to estimate precipitation amount. Temperature, cloud cover, and wind
speed are calculated following (Richardson, 1981), using cross correlation and depending on the wet/dry-state of the day. We

---

**Algorithm 1** Basic workflow of GWGEN

---

**Require:** monthly precipitation $P_{\text{in}}$ [mm], cloud cover fraction $c_{\text{in}}$, minimum ($T_{\text{min,in}}$ [°C]) and maximum ($T_{\text{max,in}}$ [°C]) temperature, wind speed $w_{\text{in}}$ [m/s], number of wet days $n_{\text{in}}$

**Output:** daily $P_i$ [mm/d], $c_i, T_i$ [°C], $w_i$ [m/s] and the wet/dry state $s_i \in \{0,1\}$

1: **for** month $m$ in $input$ **do**

2:     smooth the monthly data using Rymes and Myers (2001)

3:     Set $j = 0, \chi = 0$

4:     **while** $j \equiv 0$ or $\left|\sum_{d_i \in m} P_i - P_{\text{in}}\right| > \min\left(5\% \cdot P_{\text{in}}, 0.5mm\right)$ or $|n_{\text{sim}} - n_{\text{in}}| > 1$ **do**

5:         **for** day $d_i$ in $m$ **do**

6:             Calculate $p_{11}, p_{101}, p_{001}$ after equations (1) - (3) using $n$ {Precipitation occurence after Wilks (1999a)}

7:             Use the Markov chain to determine whether $d_i$ is wet ($s_i = 1$) or dry ($s_i = 0$)

8:             **if** $s_i = 1$ **then**

9:                 Calculate $\theta, \alpha$ and $\sigma$ via eq. (7)-(9) {Precipitation amount after Neykov et al. (2014)}

10:                 Draw a random number $P_i$ from the Gamma-GP distribution, eq. (6)

11:                 Set $T_{\text{min},i} = T_{\text{min,wet}}, T_{\text{max},i} = T_{\text{max,wet}}, c_i = c_{\text{wet}}, w_i = w_{\text{wet}}$ from eq. (10) and (12) and tables 1, 3

12:                 Set $\sigma_{T_{\text{min}},i} = \sigma_{T_{\text{min,wet}}}, \sigma_{T_{\text{max}},i} = \sigma_{T_{\text{max,wet}}}, \sigma_{w,i} = \sigma_{w,\text{wet}}, \sigma_{c,i} = \sigma_{c,\text{wet}}$ from eq. (11), (13) and (14) and tables 1, 2, 3

13:             **else**

14:                 Set $P_i = 0\,\text{mm/d}$

15:                 Set $T_{\text{min},i} = T_{\text{min,dry}}, T_{\text{max},i} = T_{\text{max,dry}}, c_i = c_{\text{dry}}, w_i = w_{\text{dry}}$ from eq. (10) and (12) and tables 1, 3

16:                 Set $\sigma_{T_{\text{min}},i} = \sigma_{T_{\text{min,dry}}}, \sigma_{T_{\text{max}},i} = \sigma_{T_{\text{max,dry}}}, \sigma_{w,i} = \sigma_{w,\text{dry}}, \sigma_{c,i} = \sigma_{c,\text{dry}}$ from eq. (11), (13) and (14) and tables 1, 2, 3

17:             **end if**

18:             Draw 4 normally distributed random numbers $\epsilon \in \mathbb{R}^4$ {Cross correlation after Richardson (1981)}

19:             Set the residuals $\chi_i = \left(\chi_{T_{\text{min}}} \quad \chi_{T_{\text{max}}} \quad \chi_c \quad \chi_w\right) = A\chi_{i-1} + B\epsilon \in \mathbb{R}^4$ with $A$ and $B$ from eq. (17)

20:             Calculate daily variables via

$$T_{\text{min},i} = \chi_{T_{\text{min}}} \cdot \sigma_{T_{\text{min}},i} + T_{\text{min},i} \qquad c_i = \chi_c \cdot \sigma_{c,i} + c_i$$
$$T_{\text{max},i} = \chi_{T_{\text{max}}} \cdot \sigma_{T_{\text{max}},i} + T_{\text{max},i} \qquad w_i = \left(\chi_w \cdot \sqrt{\sigma_{w,i}} + \sqrt{w_i}\right)^2$$

21:             Apply bias correction $w$ (eq. (23))

22:             $j = j + 1$

23:         **end for**

24:     **end while**

25: **end for**

---

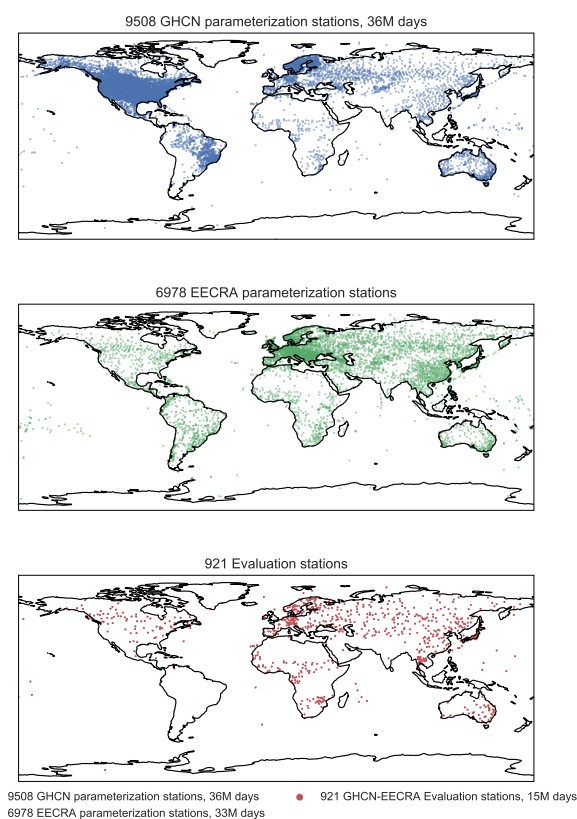

**Figure 2.** Weather stations used for parameterization and evaluation of the weather generator. The uppermost panel shows the locations of the stations used for parameterizing precipitation and temperature, the middle panel shows the stations for cloud fraction and wind speed, as well as for calculating the cross correlations between temperature, cloud fraction, and wind speed. The lower plot shows the location of the stations used to evaluate the model, which were excluded from the parameterization stations.

further add a quantile-based bias correction for wind speed and minimum temperature, which improves the simulation results significantly.

In the following subsections, we first describe the global weather station database used to develop and evaluate the model, then describe the underlying relationships that we use to define GWGEN's parameters.

## 3.1 Development of a global weather station database

To parameterize GWGEN, we assembled a global dataset of daily meteorological observations. Precipitation and minimum and maximum daily temperature come from the daily Global Historical Climatology Network (GHCN-Daily) database (Menne et al., 2012b, a). The GHCN-Daily consists of observations collected at ca. 100'000 weather stations on all continents and many oceanic islands. As the GHCN-Daily stations are highly concentrated in some parts of the world, particularly in the conterminous United States, we selected stations for our study using a geographic anti-aliasing filter to avoid an especially

strong geographic bias in the generation of the model parameters. Dividing the world up into a 0.5° grid, we selected the single station with the longest record in each cell, if one was present. While the GHCN-Daily units for precipitation have a nominal precision of 0.1 mm, several of the stations in the United States reported precipitation in fractions of an inch, which were later converted to mm. To ensure uniform precision across all of our calibration stations — this was particularly important when

generating the probability density functions for precipitation amount — we selected only those GHCN-Daily stations where all precipitation amounts between 0.1 and 1.0 mm d$^{-1}$ were reported in the record. This resulted in 9508 stations covering all continents, although the distribution is strongly heterogenous, with the majority of the stations in North America, despite our geographic filter (Figure 2, top panel). For cloud cover, windspeed, and to calculate cross-correlations between temperature, cloud cover, and windspeed, we used the Extended Edited Cloud Report Archive (EECRA) database (Hahn and Warren, 1999).

The geographic distribution of the 6978 EECRA stations we selected is different than the GHCN-Daily, with more stations in Europe (Figure 2, middle panel), but overall a relatively similar number of stations were used from both datasets. For the observations from both GHCN-Daily and EECRA, we made one additional filtering step, selecting only complete months, i.e., months with no days having missing observations, for further processing. In total, our database of daily meteorological observations used in the model parameterization contains ca. 69 million individual records.

Finally, we reserved some weather station records for model evaluation that were not used for model parameterization. These were individual stations, or two stations separated by a maximum distance of 1 km, where all of the daily meteorological variables that GWGEN simulates ($P$, $T_{\min}$, $T_{\max}$, $c$, $w$) were recorded on the same dates in the EECRA database. This merged selection from EECRA and GHCN resulted in a set of 921 stations representing ca. 15 million daily records, with observations on all continents, although the geographic distribution is once again highly heterogenous, with a particularly high density of

stations in Japan and Germany (Figure 2, bottom panel).

### 3.2   Parameterization

### 3.2.1   Precipitation occurrence

Following Geng et al. (1986), we expect to find a good relationship between the fraction of days in a month with measurable precipitation and the probability that any given day will be wet. Following Wilks (1999a) we use a hybrid-order model that

retains first-order Markov dependence for wet spells but allows second-order dependence for dry sequences; this hybrid-order scheme has been shown to be a good compromise between performance and simplicity. To parameterize the precipitation occurrence part of the model, we thus calculated transition probabilities for a wet day being followed by a wet day ($p_{11}$), for a wet day being followed by a dry day being followed by a wet day ($p_{101}$) and for two dry days being followed by a wet day ($p_{001}$). We perform this analysis on a station and month-wise basis, i.e., we first extract each of the (complete) Januaries,

Februaries, etc. for a given station, and then merge all of the Januaries (Februaries, Marches, etc...) for this station into a single series representing each month. Merging months over several years is particularly important for stations that have relatively little precipitation in a given month; for example, it could take several years of observations to observe a single ($p_{101}$) event.

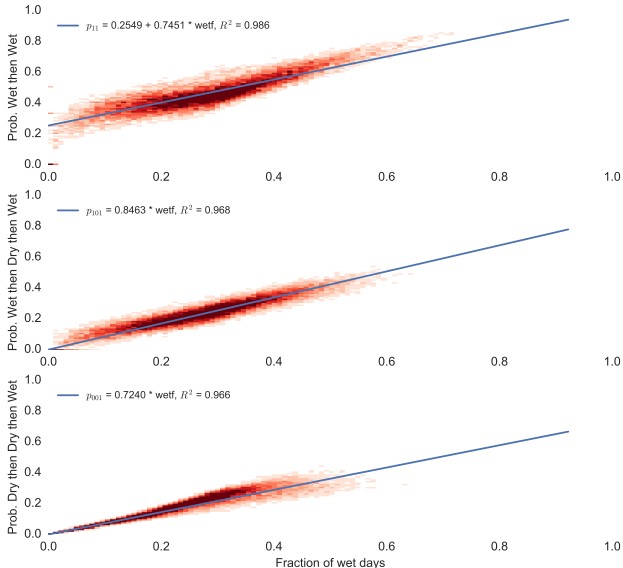

**Figure 3.** Transition probabilities vs. wet fraction. The red density plot in the background shows the density of the observations, and the blue lines are the linear regression line of the probability against the wet fraction. The fit for the $p_{11}$ transition probability was forced to the point $(1, 1)$, the others were forced to $(0, 0)$. The underlying data for the fits correspond to the means of the the multi-year series for each month for each station.

The final transition probabilities were then regressed against the fraction of days in the month with precipitation, which show the characteristic linear relationship described by Geng et al. (1986) (Figure 3).

Because the transition probabilities $(p_{001})$ and $(p_{101})$ must be zero by definition when the fraction of wet days $(f_{\mathrm{wet}})$ is zero, i.e., a completely dry month, we force the linear regression between these quantities to pass through the origin. Likewise, we require the regression line for $(p_{11})$ to equal 1 when $f_{\mathrm{wet}}$ is 1. One has to note, however, that this methodology artificially increases the $R^2$ coefficient for the fit because we fix the intercept (see for example Gordon, 1981).

The analysis results in the the following relationships:

$$p_{11} = 0.2549 + 0.7451 \cdot f_{\mathrm{wet}} \tag{1}$$

$$p_{101} = 0.8463 \cdot f_{\mathrm{wet}} \tag{2}$$

$$p_{001} = 0.7240 \cdot f_{\mathrm{wet}}. \tag{3}$$

In the weather generator (see line 6 in algorithm 1) we determine if any given day will have precipitation by calculating the appropriate probability density function selected from equations (1)-(3) on the basis of the precipitation state of the previous day (or two). Comparing the calculated probability from the selected equation with a random number $u \in [0, 1]$, a precipitation day is simulated if $u$ is greater than its corresponding probability.

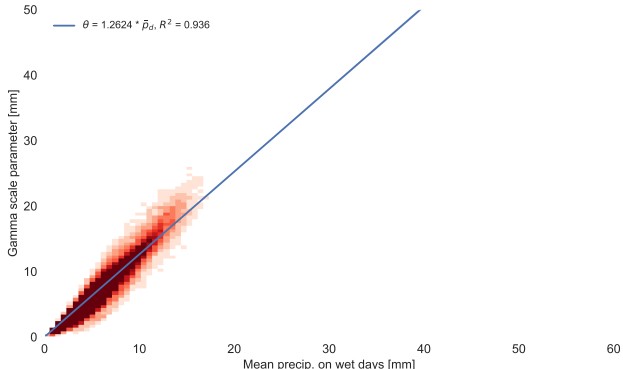

**Figure 4.** Mean precipitation - Gamma scale relationship. The blue line represents the best fit line of the mean precipitation on wet days to the estimated gamma scale parameter of the corresponding distribution. Each data point corresponds to one multi-year series of one month for one station.

### 3.2.2 Precipitation amount

Following the original WGEN (Richardson, 1981), GWGEN disaggregates precipitation amount using a statistical distribution. A number of different probability density functions have been used to estimate precipitation amount in weather generators including, e.g., single exponential or mixed exponential, one or two parameter gamma, or Weibull distribution (Wilks and

Wilby, 1999). The strong relationship between the gamma scale parameter and the mean precipitation on wet days noted by Geng et al. (1986) makes generation of precipitation amounts with only monthly input data feasible. It is based upon the fact that the expected value of a gamma random variable equals the product of its two parameters. i.e $E(\Gamma) = \alpha\theta$. The gamma distribution, however, shows poor performance in simulating high-precipitation events consistent with observations. Furrer and Katz (2008) and Neykov et al. (2014) suggest that a hybrid probability density function, based on both gamma

and the generalized pareto (GP) distribution, has superior accuracy in simulating extreme precipitation events when compared to gamma alone. Because of its superior accuracy and ease of implementation, we therefore adopt the hybrid gamma-GP distribution for simulating precipitation amount in GWGEN.

The probability density function (pdf) of the gamma distribution is defined as

$$f(x) = \begin{cases} \frac{x^{\alpha-1}e^{-\frac{x}{\theta}}}{\theta^{\alpha}\Gamma(\alpha)} & \text{for } x > 0 \\ 0 & \text{for } x = 0 \end{cases} \tag{4}$$

where $\alpha > 0$ is the shape, and $\theta > 0$ the scale parameter. The pdf of the generalized pareto (GP) distribution is defined via

$$g(x) = \begin{cases} \frac{1}{\sigma}\left(1 + \frac{\xi(x-\mu)}{\sigma}\right)^{-\frac{1}{\xi}-1} & \text{for } \xi \neq 0 \\ \frac{1}{\sigma}e^{-\frac{x-\mu}{\sigma}} & \text{for } \xi = 0 \end{cases} \tag{5}$$

with $\sigma > 0$ being the scale parameter and $\xi \in \mathbb{R}$ the shape parameter. $\mu$ is the location parameter.

Following Furrer and Katz (2008), we define the hybrid gamma-GP pdf as

$$
h(x) = \begin{cases} f(x) & \text{for } x \leq \mu, \\ (1 - F(\mu))\, g(x) & \text{for } x > \mu \end{cases},
\tag{6}
$$

where $F(\mu)$ describes the cumulative gamma distribution function at the threshold $\mu$. In our weather generator however, we first draw a random number from the gamma distribution and, if we are above the threshold, we draw another random number from the GP distribution. Thus, the frequency of precipitation events larger than $\mu$ is determined by the gamma distribution, but the actual amount of precipitation simulated when above the threshold $\mu$ is determined by the GP distribution (Furrer and Katz, 2008).

To determine the parameters of the hybrid distribution for precipitation, we started with the simple strategy by Geng et al. (1986). As above when calculating the Markov chain parameters, we created multi-year series for each of the parameterization stations for each month and extracted the days with precipitation. If a series contained more than 100 entries, we fit a gamma distribution using maximum likelihood to it in order to estimated the $\alpha$ and $\theta$ parameters.

Following Geng et al. (1986), we then fit a regression line of the gamma scale parameter against the mean precipitation on wet days $\bar{p}_d$ (see figure 4) and found the relationship

$$
\theta = 1.262\, \bar{p}_d.
\tag{7}
$$

As proposed by Geng et al. (1986), we use this relationship in our model to estimate the scale parameter of the distribution. Using this approach, the gamma shape parameter $\alpha$ is a constant, given via

$$
\alpha = \frac{\bar{p}_d}{\theta} = \frac{1}{1.262}.
\tag{8}
$$

The GP scale parameter $\sigma$ on the other hand is calculated during the simulation following Neykov et al. (2014) via

$$
\sigma = \frac{1 - F(\mu)}{f(\mu)}.
\tag{9}
$$

The other parameters of the GP distribution are obtained through a sensitivity analysis described in section 3.5.

### 3.2.3 Temperature

Following the standard WGEN methodology (Richardson, 1981) and Geng et al. (1986), daily temperature is determined through 2 processes: First, the wet/dry state of the day, and second the cross correlation (subsubsection 3.2.6).

In the weather generator, we know from the Markov chain (subsubsection 3.2.1), whether the current simulated day is a wet or dry day. Based upon the simple linear relationships

$$
\bar{x}_{\text{wet}} = c_{0,x,\text{wet}} + c_{1,x,\text{wet}} \cdot \bar{x}
$$
$$
\bar{x}_{\text{dry}} = c_{0,x,\text{dry}} + c_{1,x,\text{dry}} \cdot \bar{x}
\tag{10}
$$

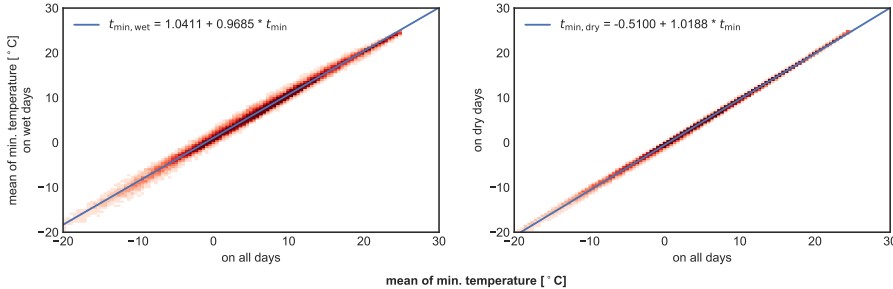

**Figure 5.** Correlation of minimum temperature on wet and dry days to the monthly mean. The y-axes show the mean minimum temperature on wet or dry days respectively, the blue line corresponds to the best fit line. Parameters of the fits are also shown in table 1.

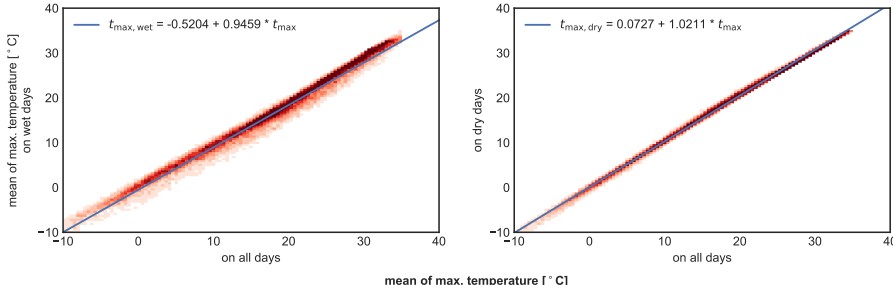

**Figure 6.** Correlation of maximum temperature on wet and dry days to the monthly mean. The y-axes show the mean maximum temperature on wet or dry days respectively, the blue line corresponds to the best fit line. Parameters of the fits are also shown in table 1.

we adjust the monthly mean $\bar{x}$ of the variable $x \in \{T_{\min}, T_{\max}\}$.

To estimate the values of the parameters $c_0$ and $c_1$ in the above equations, we follow the same procedure as for the parameters of the Markov chain (subsubsection 3.2.1). We extracted the complete months for $T_{\min}$ and $T_{\max}$ from the GHCN-Daily dataset and created a multi-year series for each month and station. We then regressed the mean on wet and dry days separated against the overall mean of each month (Figures 5 and 6). Through this procedure, we estimate the parameters necessary for equations (10) (see table 1).

To estimate residual noise, we also need an estimate of the standard deviation of the variable (see subsubsection 3.2.6). Figure 7 shows the correlation between standard deviation on wet and dry days and the corresponding mean. The means of the standard deviations (black bars in figure 7) indicate a strong but non-linear relationship between the standard deviation and the corresponding mean. The correlation changes particularly at $0°C$. We therefore use two different polynomials of order 5 for the values below and above the freezing point. Furthermore, to account for the sparse data below $-40°C$ and above $25°C$ for minimum temperature (or $-30°C$ and $35°C$ for maximum temperature), we use an extrapolation for the extremes as indicated by the blue and violet lines in figure 7. The formulae for the standard deviations $\sigma$ of minimum and maximum temperature are therefore a combination of 4 polynomials:

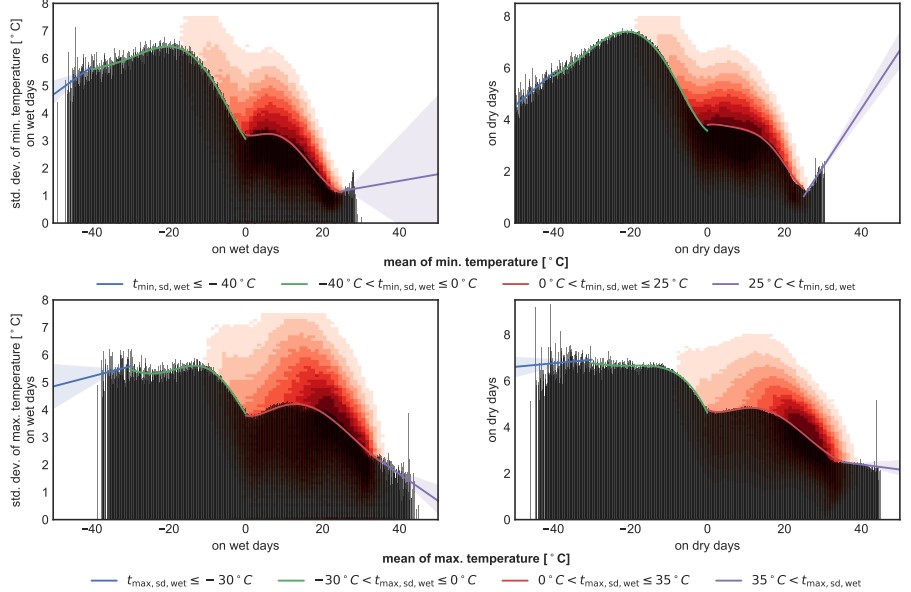

**Figure 7.** Correlation of standard deviation of the minimum and maximum temperature on wet and dry days to the monthly mean. The y-axes show the standard deviation, the x-axes the mean on wet or dry days respectively. The bars have a width of $0.1°C$ (the data accuracy) and indicate the mean standard deviation for a given mean minimum temperature in one month. The lines are fitted to these bars, where the green and red polynomials of order 5 are the use all the data below or above $0°C$ respectively and the blue and violet lines are a linear extrapolation of the data below $-40°C$ (or $-30°C$ for $T_{\max}$) or above $25°C$ (or $35°C$) respectively. The red density plot in the background indicates the spread of the data. The bars and the density plot are based on the single month for each station (i.e. not the multi-year monthly series as for, e.g. mean temperature (figure 5 and 6)). Parameters of the fits are also shown in table 1.

**Table 1.** Fit results of temperature correlation for wet and dry days for figures 5, 6, 10 and 11. The coefficients $c_0$ to $c_3$ correspond to the coefficients used in equations (10) and (14).

| plot | variable | $R^2$ | $c_0$ | $c_1$ | $c_2$ | $c_3$ |
|------|----------|-------|-------|-------|-------|-------|
| 6 | $T_{\max,\mathrm{dry}}$ | 0.9969 | 0.0727 | 1.0211 | 0 | 0 |
| 6 | $T_{\max,\mathrm{wet}}$ | 0.9752 | -0.5204 | 0.9459 | 0 | 0 |
| 5 | $T_{\min,\mathrm{dry}}$ | 0.9972 | -0.5100 | 1.0188 | 0 | 0 |
| 5 | $T_{\min,\mathrm{wet}}$ | 0.9840 | 1.0411 | 0.9685 | 0 | 0 |
| 11 | $w_{\mathrm{sd,dry}}$ | 0.4243 | 0 | 1.0860 | -0.2407 | 0.0222 |
| 11 | $w_{\mathrm{sd,wet}}$ | 0.5003 | 0 | 0.8184 | -0.1263 | 0.0093 |
| 10 | $w_{\mathrm{dry}}$ | 0.9930 | 0 | 0.9437 | 0 | 0 |
| 10 | $w_{\mathrm{wet}}$ | 0.9723 | 0 | 1.0937 | 0 | 0 |

**Table 2.** Fit results of the correlation of temperature standard deviation with the corresponding mean on wet/dry days for figure 7. The underlying equations are shown in equation (11).

| variable | interval | $R^2$ | $c_0$ | $c_1$ | $c_2$ | $c_3$ | $c_4$ | $c_5$ |
|---|---|---|---|---|---|---|---|---|
| $T_{\max,\mathrm{sd,dry}}$ | (-∞, -30] | 0.0125 | 7.3746 | 0.0154 | 0 | 0 | 0 | 0 |
| | (-30, 0.0] | 0.6721 | 4.6170 | -0.3387 | -0.0188 | -0.0003 | 0.000003 | 0.0000001 |
| | (0.0, 35] | 0.9744 | 4.7455 | -0.0761 | 0.0189 | -0.0013 | 0.00003 | -0.0000002 |
| | (35, ∞) | 0.0390 | 3.2554 | -0.0218 | 0 | 0 | 0 | 0 |
| $T_{\max,\mathrm{sd,wet}}$ | (-∞, -30] | 0.0366 | 6.6720 | 0.0364 | 0 | 0 | 0 | 0 |
| | (-30, 0.0] | 0.7362 | 3.8601 | -0.2186 | 0.0039 | 0.0015 | 0.00006 | 0.0000007 |
| | (0.0, 35] | 0.9508 | 3.7919 | -0.0313 | 0.0161 | -0.0012 | 0.00003 | -0.0000002 |
| | (35, ∞) | 0.2530 | 5.5529 | -0.0973 | 0 | 0 | 0 | 0 |
| $T_{\min,\mathrm{sd,dry}}$ | (-∞, -40] | 0.6006 | 10.8990 | 0.1271 | 0 | 0 | 0 | 0 |
| | (-40, 0.0] | 0.9509 | 3.5676 | -0.1154 | 0.0282 | 0.0020 | 0.00004 | 0.0000003 |
| | (0.0, 25] | 0.9825 | 3.7941 | 0.0330 | -0.0150 | 0.0019 | -0.0001 | 0.000002 |
| | (25, ∞) | 0.7784 | -4.6194 | 0.2261 | 0 | 0 | 0 | 0 |
| $T_{\min,\mathrm{sd,wet}}$ | (-∞, -40] | 0.1661 | 9.7272 | 0.1011 | 0 | 0 | 0 | 0 |
| | (-40, 0.0] | 0.9285 | 3.0550 | -0.2116 | 0.0137 | 0.0014 | 0.00004 | 0.0000003 |
| | (0.0, 25] | 0.9633 | 3.2187 | -0.0451 | 0.0209 | -0.0026 | 0.00010 | -0.000001 |
| | (25, ∞) | 0.0089 | 0.5571 | 0.0244 | 0 | 0 | 0 | 0 |

$$
\sigma_{T_{\min},\mathrm{wet/dry}} =
\begin{cases}
p_1(\bar{T}_{\min,\mathrm{wet/dry}}), & \text{for } \bar{T}_{\min,\mathrm{wet/dry}} \leq -40^\circ C \\
p_5(\bar{T}_{\min,\mathrm{wet/dry}}), & \text{for } -40^\circ C < \bar{T}_{\min,\mathrm{wet/dry}} \leq 0^\circ C \\
p_5(\bar{T}_{\min,\mathrm{wet/dry}}), & \text{for } 0^\circ C < \bar{T}_{\min,\mathrm{wet/dry}} \leq 25^\circ C \\
p_1(\bar{T}_{\min,\mathrm{wet/dry}}), & \text{for } 25^\circ C < \bar{T}_{\min,\mathrm{wet/dry}}
\end{cases}
$$

$$
\sigma_{T_{\max},\mathrm{wet/dry}} =
\begin{cases}
p_1(\bar{T}_{\max,\mathrm{wet/dry}}), & \text{for } \bar{T}_{\max,\mathrm{wet/dry}} \leq -30^\circ C \\
p_5(\bar{T}_{\max,\mathrm{wet/dry}}), & \text{for } -30^\circ C < \bar{T}_{\max,\mathrm{wet/dry}} \leq 0^\circ C \\
p_5(\bar{T}_{\max,\mathrm{wet/dry}}), & \text{for } 0^\circ C < \bar{T}_{\max,\mathrm{wet/dry}} \leq 35^\circ C \\
p_1(\bar{T}_{\max,\mathrm{wet/dry}}), & \text{for } 35^\circ C < \bar{T}_{\max,\mathrm{wet/dry}}
\end{cases}
\tag{11}
$$

$p_1$ in eq. (11) denotes a polynomial of order 1, $p_5$ a polynomial of order 5. The coefficients of the different polynomials are shown in table 2.

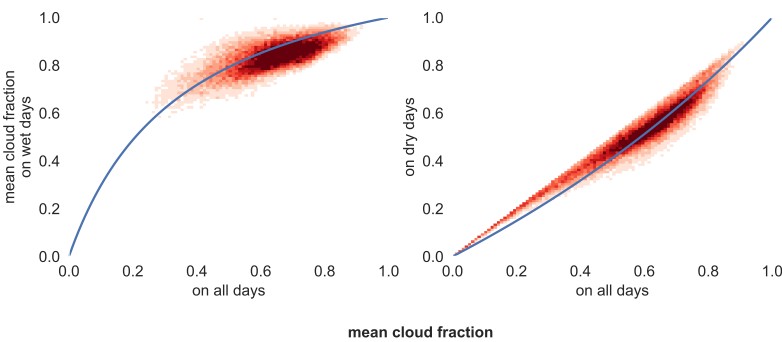

**Figure 8.** Correlation of cloud fraction on wet and dry days to the monthly mean.The y-axes show the mean cloud fraction on wet or dry days respectively, the blue line corresponds to the best fit line. Parameters of the fits are also shown in table 3.

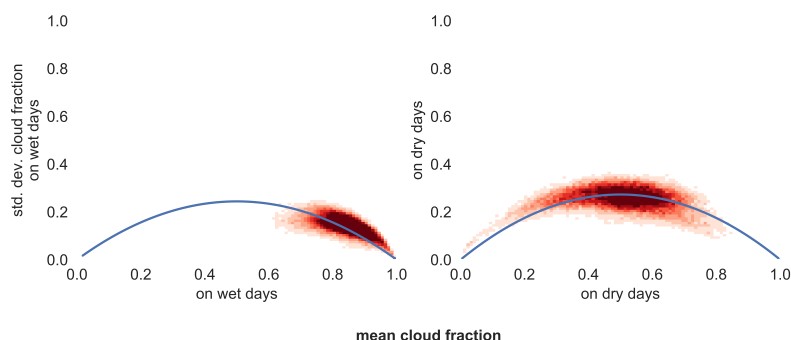

**Figure 9.** Correlation of standard deviation of the cloud fraction on wet and dry days to the corresponding monthly mean. The y-axes show the standard deviation, the x-axes the mean on wet or dry days respectively. The blue line corresponds to the best fit line. Parameters of the fits are also shown in table 3.

These coefficients are based on the means of the standard deviation (black bars in figure 7). We chose this procedure to give the same weight to all temperatures. Otherwise the fit would be dominated by the temperature values around the freezing points.

### 3.2.4 Cloud fraction

5 Monthly mean cloud fraction is disaggregated, as for temperature, using the standard WGEN procedure of adding statistical noise to a wet- or dry-day mean and accounting for cross-correlation among the different weather variables. For the parameterization of the cloud fraction equations, we used the EECRA dataset. The original dataset contains eight measurements per day of the total cloud cover in units of octas, i.e., values ranging from 0 (clear sky) to 8 (overcast). Hence, to calculate the daily cloud fraction, those values were averaged and divided by 8 to produce a daily mean.

**Table 3.** Fit results of cloud correlation for wet and dry days for figure 8

| plot | variable | $a$ | std. err. of $a$ | $R^2$ |
|------|----------|-----|------------------|-------|
| 8 | $c_{\text{dry}}$ | 0.4302 | 0.0013 | 0.8745 |
| 8 | $c_{\text{wet}}$ | -0.7376 | 0.0006 | 0.3881 |
| 9 | $c_{\text{sd,dry}}$ | 1.0448 | 0.0004 | 0.2803 |
| 9 | $c_{\text{sd,wet}}$ | 0.9881 | 0.0006 | 0.0802 |

To adjust the monthly mean depending on the wet/dry state of the day, we could not use a simple linear relationship as we used for temperature because cloud fraction is bounded by a lower limit 0 and an upper limit of 1. Furthermore, we observed that cloud cover on wet days is usually greater or equal to the monthly mean cloud cover, whereas the cloud cover on dry days is usually less or equal to the monthly mean cloud cover. This results in a concave curve for the wet case and a convex curve

for dry days. We used a qualitative graphical analysis to develop "best guess" equations that had the desired shape and propose the following formulae for the regression linking cloud cover on wet or dry days to the overall mean:

$$
\bar{c}_{\text{wet}} = \frac{-a_{\text{c,wet}} - 1}{a_{\text{c,wet}}^2 \cdot \bar{c} - a_{\text{c,wet}}^2 - a_{\text{c,wet}}} - \frac{1}{a_{\text{c,wet}}}
$$
$$
\bar{c}_{\text{dry}} = \frac{-a_{\text{c,dry}} - 1}{a_{\text{c,dry}}^2 \cdot \bar{c} - a_{\text{c,dry}}^2 - a_{\text{c,dry}}} - \frac{1}{a_{\text{c,dry}}} \tag{12}
$$

with $a_{\text{c,wet}} < 0$ and $a_{\text{c,dry}} > 0$.

The standard deviation of cloud cover fraction becomes 0 when the mean monthly cloud fraction reaches both the minimum or maximum limits of 0 and 1. Hence, for $c_{\text{sd,dry}}$ and $c_{\text{sd,wet}}$ we have an concave parabola with the formula

$$
\sigma_{\text{c,wet}} = a_{\text{c,wet}}^2 \cdot \bar{c}_{\text{wet}} \cdot (1 - \bar{c}_{\text{wet}})
$$
$$
\sigma_{\text{c,dry}} = a_{\text{c,dry}}^2 \cdot \bar{c}_{\text{dry}} \cdot (1 - \bar{c}_{\text{dry}}) \tag{13}
$$

with $a_{\text{c,wet}}, a_{\text{c,dry}} \geq 0$. Results of the fits can be seen in figure 8, 9 and the parameters in table 3.

### 3.2.5 Wind speed

The parameterization of the mean wind speed is based upon the same linear equation (10) as temperature. For the standard deviation however, we use a third-order polynomial given that is forced through the origin, given via

$$
\sigma_{w,\text{wet}}(\bar{w}_{\text{wet}}) = c_{1,w,\text{wet}} \, \bar{w}_{\text{wet}} + c_{2,w,\text{wet}} \, \bar{w}_{\text{wet}}^2 + c_{3,w,\text{wet}} \, \bar{w}_{\text{wet}}^3
$$
$$
\sigma_{w,\text{dry}}(\bar{w}_{\text{dry}}) = c_{1,w,\text{dry}} \, \bar{w}_{\text{dry}} + c_{2,w,\text{dry}} \, \bar{w}_{\text{dry}}^2 + c_{3,w,\text{dry}} \, \bar{w}_{\text{dry}}^3. \tag{14}
$$

This better resolves the complex behavior close to $0 \, \text{m s}^{-1}$ compared to a linear fit. The plots are shown in the figures 10 and 11 and the parameters for the fits are shown in table 1.

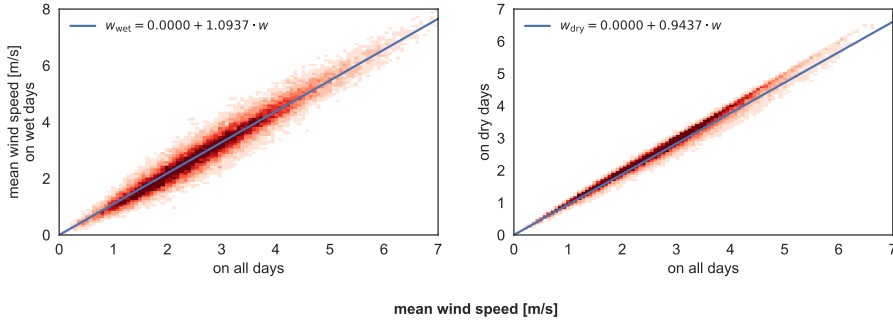

**Figure 10.** Correlation of wind speed on wet and dry days to the monthly mean. The y-axes show the mean cloud fraction on wet or dry days respectively, the blue line corresponds to the best fit line. Parameters of the fits are also shown in table 1.

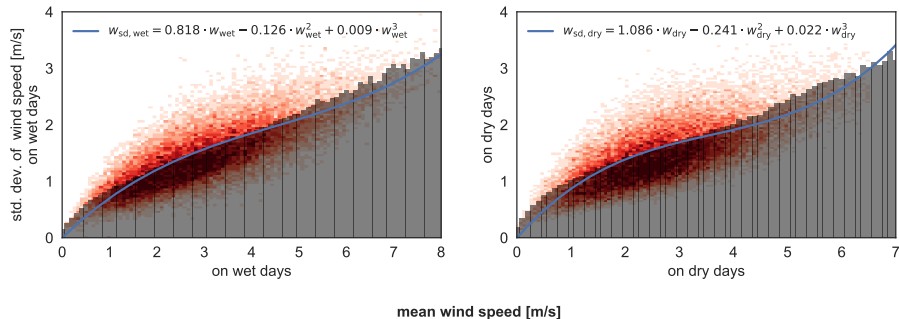

**Figure 11.** Correlation of standard deviation of wind speed on wet and dry days to the corresponding monthly mean. The y-axes show the standard deviation, the x-axes the mean on wet or dry days respectively. The blue line corresponds to the best fit line, a third order polynomial to the underlying red density plot. The black bars have a width of $0.1\,\mathrm{m\,s^{-1}}$, the accuracy of the input data, and indicate the mean standard deviations for the given interval range. Parameters of the fits are also shown in table 1.

### 3.2.6 Cross correlation

Following Richardson (1981) we use cross correlation to add additional residual noise to the simulated meteorological variables, which provides more realism in the daily weather result. This methodology, based on Matalas (1967) preserves the serial and the cross correlation between the simulated variables. It implies that the serial correlation of each variable may be described by a first-order linear autoregressive model

Given the cross correlation matrix $M_0 \in \mathbb{R}^4 \times \mathbb{R}^4$ and the lag-1 correlation matrix $M_1 \in \mathbb{R}^4 \times \mathbb{R}^4$, we calculate

$$A = M_1 M_0^{-1} \qquad BB^T = M_0 - M_1 M_0^{-1} M_1^T. \tag{15}$$

The matrices $A, B, M_0$ and $M_1$ are calculated using the stations from the EECRA database in figure 2. The results are

$$M_0 = \begin{pmatrix} 1. & 0.565 & 0.041 & 0.035 \\ 0.565 & 1. & -0.089 & -0.043 \\ 0.041 & -0.089 & 1. & 0.114 \\ 0.035 & -0.043 & 0.114 & 1. \end{pmatrix} \qquad M_1 = \begin{pmatrix} 0.933 & 0.55 & 0.016 & 0.03 \\ 0.557 & 0.417 & -0.066 & -0.043 \\ 0.004 & -0.095 & 0.599 & 0.093 \\ 0.011 & -0.063 & 0.061 & 0.672 \end{pmatrix}. \tag{16}$$

leading to

$$A = \begin{pmatrix} 0.916 & 0.031 & -0.018 & 0.001 \\ 0.485 & 0.135 & -0.069 & -0.047 \\ 0.004 & -0.043 & 0.592 & 0.023 \\ 0.012 & -0.043 & -0.02 & 0.672 \end{pmatrix} \qquad B = \begin{pmatrix} 0.358 & 0. & 0. & 0. \\ 0.112 & 0.809 & 0. & 0. \\ 0.142 & -0.06 & 0.785 & 0. \\ 0.077 & -0.016 & 0.061 & 0.733 \end{pmatrix}. \tag{17}$$

The columns and rows in the two matrices correspond to min. and max. temperature, cloud fraction and square root of wind speed, respectively.

In the weather generator, the variables $T_{\min}, T_{\max}, c$ and $w$ are then calculated using a combination of residual noise $\chi_i$ (where $i$ denotes the current simulated day) and the mean of the variables. $\chi_i$ is determined by the other variables and the previous day using $A$ and $B$ from above (Richardson, 1981; Matalas, 1967). Hence, $\chi_i$ is given via

$$\chi_i = \begin{pmatrix} \chi_{T_{\min}} & \chi_{T_{\max}} & \chi_c & \chi_w \end{pmatrix} = A\chi_{i-1} + B\epsilon \in \mathbb{R}^4. \tag{18}$$

The daily values for the variables are then calculated via

$$T_{\min,i} = \chi_{T_{\min}} \cdot \sigma_{T_{\min},\text{wet/dry}} + \bar{T}_{\min,\text{wet/dry}} \qquad c_i = \chi_c \cdot \sigma_{c,\text{wet/dry}} + \bar{c}_{\text{wet/dry}} \tag{19}$$

$$T_{\max,i} = \chi_{T_{\max}} \cdot \sigma_{T_{\max},\text{wet/dry}} + \bar{T}_{\max,\text{wet/dry}} \qquad w_i = \left( \chi_w \cdot \sqrt{\sigma_{w,\text{wet/dry}}} + \sqrt{\bar{w}_{\text{wet/dry}}} \right)^2 \tag{20}$$

with $\sigma_{T_{\min},\text{wet/dry}}, \sigma_{T_{\max},\text{wet/dry}}$ from eq. (11), $\sigma_{c,\text{wet/dry}}$ from eq. (13), $\sigma_{w,\text{wet/dry}}$ from eq. (14), $\bar{T}_{\min,\text{wet/dry}}, \bar{T}_{\max,\text{wet/dry}}, \bar{w}_{\text{wet/dry}}$ from eq. (10) and $\bar{c}_{\text{wet/dry}}$ from eq. (12).

Since this procedure always requires the residuals from the previous day, $\chi_{i-1}$, we initialize $\chi_0$ with 0, simulate the month and then simulate it again.

Note that, through the entire procedure, wind speed is subject to a square-root transformation (also when calculating $M_0$ and $M_1$) to account for the fact that it is not normally distributed.

## 3.3 Model Evaluation

To evaluate GWGEN, we started with the daily meteorology at the evaluation stations described above and calculated monthly summaries. We used this monthly data to drive the model and simulate daily meteorology. The resulting daily series now has

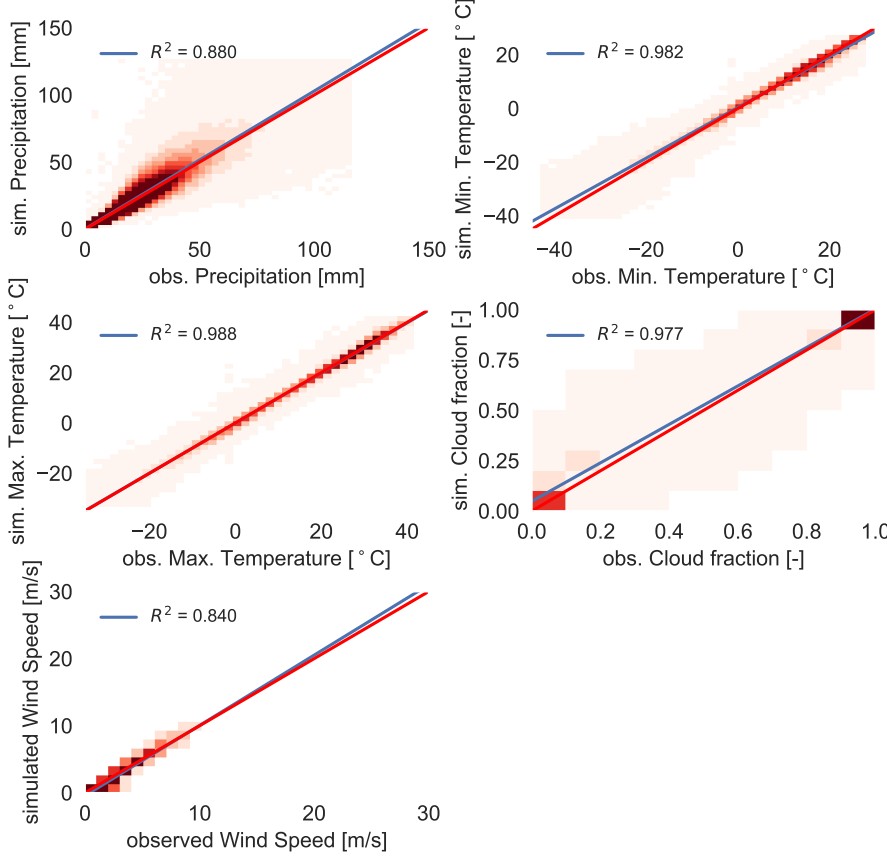

**Figure 12.** QQ-plots for all variables with all quantiles (1, 5, 10, 25, 50, 75, 90, 95 and 99) for $\mu = 5.0\,\mathrm{mm}\,\mathrm{mm}$, $\xi = 1.5$. The blue lines are linear regression from simulation to observation. The red line shows the ideal fit (the identity line). Blue shaded areas represent the $95\%$ confidence interval. The plots compares the simulated quantile from the list above of one year of one station to the corresponding observed quantile of the same year and station. The plot for wind speed underwent used the bias correction from subsection 3.4.

the same length as the observed meteorology from the GHCN and EECRA database. Because we cannot expect the weather generator to reproduce the weather exactly as observed, for example the number of rainy days in a month may be the same as observed but they may not occur in precisely the same order, our evaluation is restricted to comparing the statistical properties of the input observed versus the output simulated daily meteorology.

5    Figure 12 shows the comparison of simulated versus observed values for each of the five meteorological variables handled by GWGEN. For temperature, wind, and cloud fraction, the model does an excellent job of downscaling monthly input to daily resolution[1]. The comparison between precipitation amounts looks good when considering all of the data, however a closer look into the results (Fig. 13) shows that while the higher precipitation percentiles are well captured using the hybrid Gamma-GP

---

[1]Note that the plot for wind speed has been bias corrected using the approach in subsection 3.4.

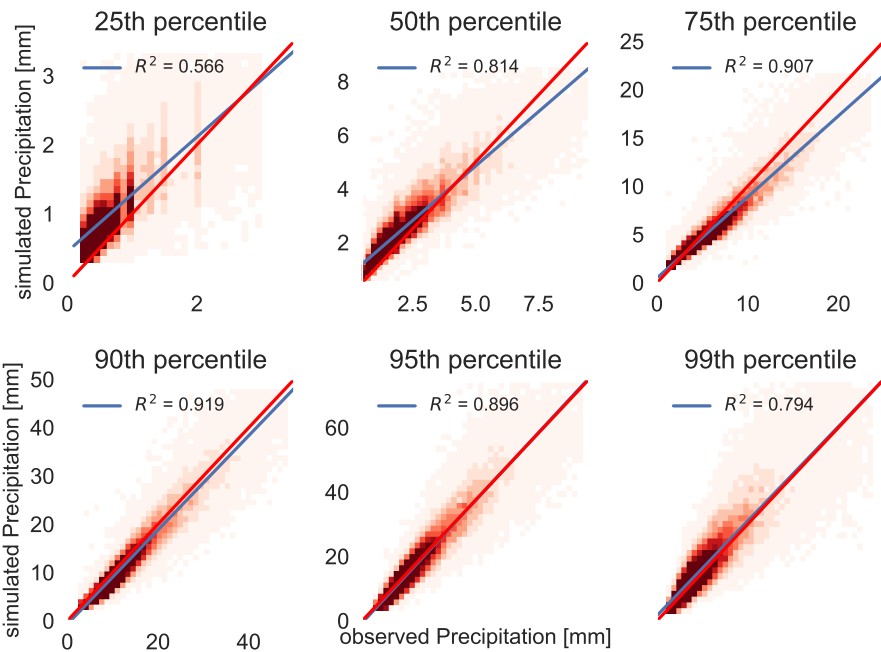

**Figure 13.** QQ-plot for different quantiles for precipitation for $\mu = 5.0\text{mm}$, $\xi = 1.5$. The blue lines are linear regression from simulation to observation. The red line shows the ideal fit (the identity line). Blue shaded areas represent the $95\%$ confidence interval. The plots compare the simulated quantile of one year of one station to the corresponding observed quantile of the same year and station.

distribution, the lower percentiles show somewhat worse results. This observation of poor performance for very low values also holds true for wind speed (not shown here). The lower values of the two variables, however, are very close to the precision of the observation ($0.1\,\text{mm}$ for precipitation and $0.1\,\text{m}\,\text{s}^{-1}$ for wind speed). Very small precipitation amounts and low wind speeds are also less biophysically and ecologically important compared to the higher percentiles. We therefore consider the results of the evaluation largely acceptable.

In table 4 we also compare the simulated versus the observed frequencies. For very light rain (<=1mm), light rain (1-10mm), heavy rain (10-20mm) and very heavy rain (>20mm). As we can see, our model underestimates the occurrence of very light rain events ($28.6\%$ instead of $36.4\%$) and overestimates the light rain events ($58.3\%$ instead of $48.6\%$) but generally performs much better than GCMs (Dai, 2006; Sun et al., 2006), especially when it comes to the heavy rain events.

## 3.4 Bias correction

After evaluating the results of GWGEN for wind speed for the different quantiles (see previous subsection 3.3) we found a strong, systematic bias between the simulated and the observed values. This observation led us to adopt a further measure to improve the quality of the model output by implementing a quantile-based bias correction.

**Table 4.** Simulated and observed precipitation frequencies for certain ranges. The frequency is defined as the number of precipitation occurences in the specified range, divided by the total number of precipitation occurences.

| Precip. range [mm] | Simulated | Observed |
|---|---|---|
| (0, 1] | 0.285688 | 0.364014 |
| (1, 10] | 0.583330 | 0.486415 |
| (10, 20] | 0.074063 | 0.090178 |
| (20, ∞] | 0.056920 | 0.059392 |

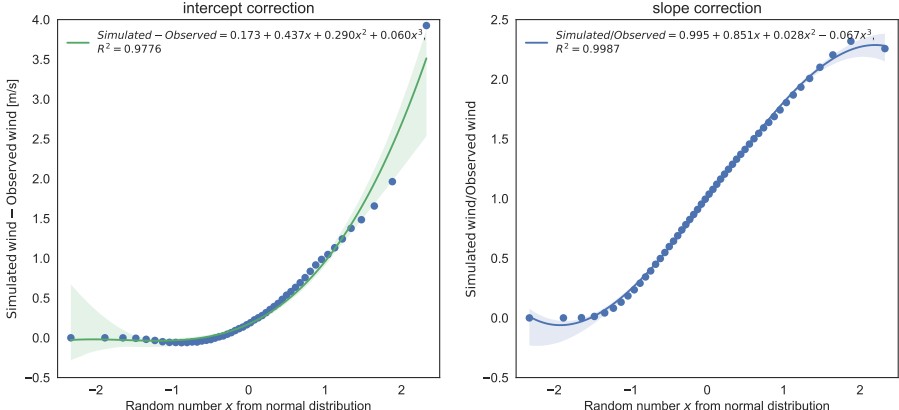

**Figure 14.** Basis for the wind bias correction. For the left plot, each data point corresponds to the difference of a simulated percentile to the observed percentile. For the right plot (wind speed), each data point corresponds to the fraction of simulated to the observed wind speed for a given percentile. The random number on the x-axis represents the residual value from a normal distribution centered at 0 with standard deviation of unity, as it is used in the cross correlation approach (Richardson, 1981).

We use an empirical distribution correction approach (quantile-mapping) (Lafon et al., 2012) to a posteriori correct the simulated data. In the quantile evaluation (previous subsection 3.3) we saw that the simulated wind speed is a linear function of the observed wind speed, i.e. $w_{sim} = \text{intercept} + \text{slope} \cdot w_{obs}$ (best fit line in figure 12). Therefore, we use two steps here, one is for the difference between simulation and observation (ideally 0), the other one is the fraction of observation and simulation (ideally 1). The first one corresponds to the intercept with the y-axis in figure 12, the second one to the slope of the best fit line. The analysis is based on every second percentile between 1 and 100 (i.e. $1, 3, 5, \ldots$) and mapped to it's corresponding random number $u \in \mathbb{R}$ from a normal distribution as it is used for the cross correlation in the weather generator (section 3.2.6, x-axis in figure 14 and Richardson (1981)).

Regarding the intercept (fig. 14, left) we see that it strongly follows an exponential function given through

$$f_{exp}(u) = e^{a\,u+b}, \qquad a, b, u \in \mathbb{R}. \tag{21}$$

The slope (fig. 14, right) on the other hand can be described by a simple third-order polynomial given by

$$p3(u) = c_0 + c_1 u + c_2 u^2 + c_3 u^3, \qquad c_0, c_1, c_2, c3, u \in \mathbb{R} \tag{22}$$

Hence, given the best fit lines in figure 14, the simulated wind speed is corrected via

$$w'_{sim} = \frac{w_{sim} - f_{exp}(u)}{p3(u)} \tag{23}$$

with $a = 1.1582, b = -1.3359, c_0 = 0.9954, c_1 = 0.8508, c_2 = 0.0278, c_3 = -0.0671$.

## 3.5  Sensitivity analysis

The Generalized-Pareto part of the hybrid Gamma-GP distribution, which we used to simulate precipitation amount, has two parameters: the GP shape, and the threshold parameter. Unlike the gamma parameters, we were unable to relate these GP parameters to any of the monthly summary data we use as input to GWGEN. Hence, we decided to set fixed values for these

parameters, and determine them through a sensitivity analysis.

To select the "best" values of the GP parameters, we compared simulated with observed precipitation amounts, running GWGEN with a wide range of realistic parameter values. To quantitatively assess the model performance, we used two metrics: 1) direct comparison of the quantiles (see previous section), and 2) a Kolmogorov-Smirnov (KS) test that evaluates whether two data samples come from significantly different distributions. Our criteria were

1. The $R^2$ correlation coefficient between simulated and observed quantiles

2. The fraction $\frac{\text{simulated precipitation}}{\text{observed precipitation}}$ from the slopes in figure 13 and it's deviation from unity

3. the fraction of simulated (station specific) years that are significantly different (KS test) from the observation

4. The mean of the above values

We tried two different approaches to select the gamma-GP crossover threshold: first we tried a fixed crossover point, second

we used a quantile-based crossover point. For the latter, the model chooses to use the GP distribution if the quantile of the random number drawn from the gamma distribution is above a certain quantile threshold. This introduces a flexible crossover point in our hybrid distribution which, however, did not improve the results significantly. We therefore show here only the results using the fixed crossover point.

The values of the crossover point for our sensitivity analysis were 2, 2.5, 3, 4 and from 5 to 20 in steps of 2.5 and 20 to

100 in steps of 5. Furthermore we varied the GP shape parameter from 0.1 to 3 in steps of 0.1 (810 experiments in total). The results of this sensitivity analysis are shown in the supplementary material, figure 15.

In general we found that the three criteria 1, 2 and 3 could not be optimized all together at the same time. The $R^2$ is best for high thresholds and low GP shape parameters, the slope is best for low to intermediate thresholds and a low GP shape and the KS statistic is best for low threshold and intermediate GP shape parameters.

However, $R^2$ did not vary that much (from 0.68 to 0.74) and from a visual evaluation of the corresponding quantile plots we saw that the higher quantiles (>90) were much better represented for a better KS result. Hence we chose to follow the KS test criteria, which is also the strictest of our evaluation methods but again compared the different quantile plots to get good results for the higher quantiles. Finally, we chose a threshold of $5\,\text{mm}$ and a GP shape parameter of $1.5$. For this setting, $81.7\%$ of the simulated years do not show a significant difference compared to the observation, the mean $R^2$ of the plots in figure 13 is $0.81$ and the mean deviation of the slope from unity is $0.10$ and for the upper quantiles (90 to 100), $0.017$.

Nevertheless, in total the results seem to be fairly independent of the two parameters since even the amount of years without significant differences vary from $73\%$ to only $83\%$. It is however better than the gamma distribution alone which still has $78.6\%$ of station years not differing significantly but with a slope deviation from unity for the upper quantiles of $0.16$. Thus using the hybrid Gamma-GP distribution improves the simulation of high-amount precipitation events by roughly factor 10 compared to a standard Gamma approach.

## 4   Limitations

As demonstrated above, GWGEN successfully downscales monthly to daily meteorology with good correlation and low bias when compared to observations. However, there are a few limitations of the model as currently described that should be noted. Importantly, this version of GWGEN neither downscales all conceivable meteorological variables, nor does it provide a mechanism for generating daily meteorological timeseries across multiple points that are spatially autocorrelated. Concerning the former point, while GWGEN simulates daily precipitation, temperature, cloud cover, and windspeed, it does not currently handle other variables that might be important in land surface modeling, such as humidity or wind direction. On the latter point, the lack of explicit simulation of spatial autocorrelation may make GWGEN unsuitable for certain applications, e.g., regional high-resolution hydrological modeling in small catchments ($<$ ca. 2500 km$^2$), where having the capability to simulate flood and other extremes is important. This is because the the weather generator could, e.g., simulate rainfall on different days in different parts of the catchment, where in reality storm events would be highly autocorrelated in space and controlled by mesoscale meteorological conditions.

## 5   Discussion and Outlook

GWGEN successfully downscales monthly to daily meteorology, for any point on the globe, in any climate, in any season, and in any time in recent earth history and into the near future (e.g., next century). It extends the original Richardson-type weather generators to simulate wind speed along with precipitation, temperature, and cloud cover. The model requires only monthly values of the meteorological variables to be downscaled, and does not rely on any other spatial information, e.g., whether or not the location is in the tropics.

In general, the results of our downscaled meteorology are excellent, with all simulated variables showing both very high correlation and limited bias when compared to observations. We improved the simulation of daily precipitation amount by

replacing the Gamma distribution used in the original Richardson-type weather generators with a hybrid Gamma-GP distribution, which results in the improved simulation of heavy precipitation events. The GP distribution is based upon a globally fixed shape and location parameter, which may be an oversimplification, but is still ten times more accurate than traditional methods that used Gamma alone. Our extensive sensitivity analysis to determine the best coefficients for the shape and location parame-

ters of the GP distribution suggest that further improvements might come through correlating the GP parameters to geographic region and/or seasonality (Maraun et al., 2009; Rust et al., 2009) or by introducing a dynamical location parameter (Frigessi et al., 2002). Finally, we introduced a step to correct for systematic bias in the downscaling of temperature and wind speed.

Despite the limitations noted above, GWGEN will be useful in a wide range of applications, from global vegetation and crop modeling, to large-scale hydrologic analyses, to understanding animal behavior, to forecasting of fire, insect outbreaks,

and other ecosystem disturbances. GWGEN may even be envisaged as a potential replacement for very large and cumbersome gridded datasets of high-temporal resolution meteorology such as CRUNCEP (Viovy and Ciais, 2016), especially for models that use meteorological forcing at a daily timestep. The weather generator is particularly suited for the incorporation into models that run on a spatial grid, for example, GWGEN can readily be incorporated into existing DGVMs such as LPJ-LMfire (Pfeiffer et al., 2013) or LPJ-ML (Bondeau et al., 2007) that already rely on a weather generator to provide daily meteorology

for certain processes.

While GWGEN does not handle spatial autocorrelation, in most DGVMs there is no lateral connection between gridcells, and therefore an explicit representation of spatial autocorrelation in the driving daily meteorological data would have no effect on the model output. We further note that if the monthly data used to drive the model are spatially autocorrelated — this would be the case when using gridded climatology for example — then the result of the weather generator will also preserve this

autocorrelation, at least when the model results are analyzed on monthly or longer timescales.

The limitations present in this version of GWGEN could be addressed in future versions. Methods for simultaneous multisite weather generation exist (Wilks, 1998, 1999b, c) and could be adapted to GWGEN. However, even simpler methods to approximate spatial autocorrelation could be possible. Running GWGEN with gridded monthly meteorology — this is the primary application we foresee for the current version of the model — means that the input variables are already highly correlated

in space, i.e., the monthly climate in one gridcell generally closely resembles neighboring cells, outside of complex terrain containing sharp, monotonic climate gradients, e.g., rain shadows. Thus, one simple way of achieving a measure of spatial autocorrelation in GWGEN would be to impose a spatial autocorrelation field on the sequence of random numbers used to impose stochastic noise in the downscaling functions. If the random number sequence is similar between gridcells, then, e.g., rain is likely to fall on the same day, given that the transition probabilities will likely also be similar. Over moderate distances,

e.g., <50's of km, it might even be sufficient to use the same random seed across all gridcells in a neighborhood. This would have the effect of producing strongly autocorrelated daily meteorology in space, with the only variations being imposed by the underlying input monthly climatology.

Furthermore, it would be straightforward to include additional meteorological variables in the model framework, handling, e.g., humidity in the same way that temperatures, cloud cover, and wind speed are disaggregated. Other variables, such as

pressure and wind direction, might be more difficult using the basic GWGEN structure because of the importance of auto-

correlation, particularly at high spatial resolution, and might benefit from a different approach towards weather generation. Finally, GWGEN only downscales meteorology from monthly to daily values; for models that require an even shorter timestep, e.g., 6-hourly, some extension of the model functionality would be required. For certain variables, e.g., temperatures, sub-daily downscaling could be easily implemented (Cesaraccio et al., 2001), for other variables, such as precipitation, a large literature on downscaling methods exists (e.g. Bennett et al., 2016), and global datasets of hourly meteorology for model calibration are available (e.g., the Integrated Surface Database, Smith et al., 2011).

## 6    Conclusions

Compiling a global database of daily precipitation, temperature, cloud cover, and wind speed measurements, we explored the relationship between daily meteorology and monthly summaries first described in the context of weather downscaling by Geng and Auburn (1987). Our analysis of more than 50 million individual records showed that daily-to-monthly relationships are relatively stable in space and time, and constant across a very wide range of stations from all latitudes and climate zones. With the resulting relationships, we parameterized a WGEN/SIMMETEO-type weather generator, with the intention of creating a generic scheme that could be applied anywhere over the earth's land surface for the past, present, and (near) future.

## 7    Code availability

GWGEN, is open source software, and the code, utility programs for parameterization, evaluation and manipulating the raw weather station data, and complete documentation are available at (Sommer and Kaplan, 2017). The original weather station database can be made available upon request to the authors or downloaded from Hahn and Warren (1999) and Menne et al. (2012b). The weather generator module is programmed in FORTRAN, the parameterization, evaluation and other supplementary tools are written in Python mainly using the numerical python libraries numpy and scipy (Jones et al., 2001), statsmodels (Seabold and Perktold, 2010), as well as matplotlib (Hunter, 2007) and psyplot (Sommer, 2017) for the visualization. Detailed installation instructions can be found in the user manual: https://arve-research.github.io/gwgen/.

# Appendix A:  Supplementary material

## A1    Sensitivity analysis

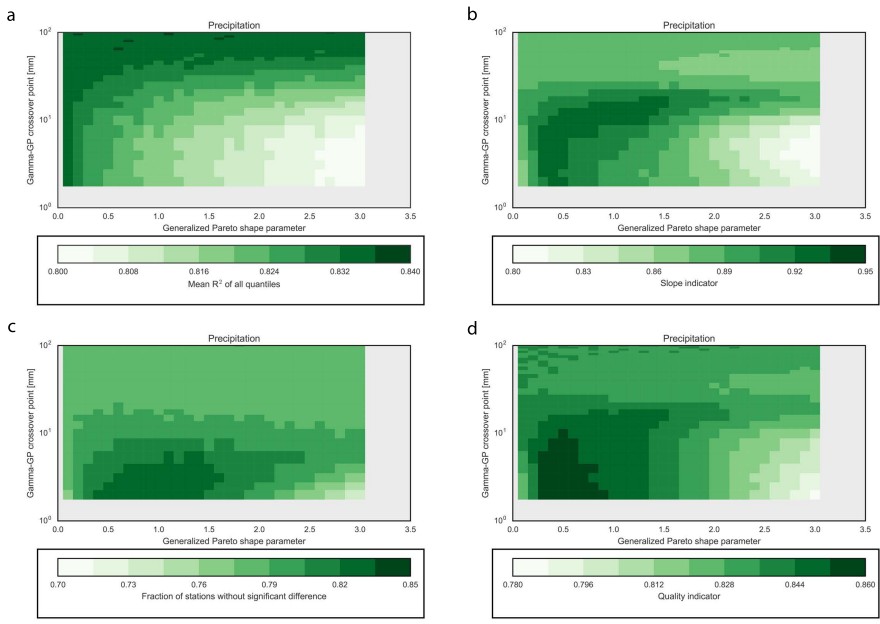

**Figure 15.** Results of the sensitivity analysis for the (a) correlation coefficient $R^2$, (b) deviation from a slope of unity, (c) the fraction of significant different station years, (d) the mean of (a) - (c). For the plots in (a) and (b) we used the means of the 25th, 50th, 75th, 90th, 95th and 99th percentiles. In general, 1 (dark green) is best, 0 (white) is worst. The dark red fields indicate experiments that failed because of a too low threshold and too high GP shape parameter. Note the logarithmic scale on the y-axis.

*Author contributions.*  JOK conceived the model and analyses, wrote the prototype code and performed preliminary analyses, PS developed and documented the final version of the code (including parameterization and evaluation), performed all of the final analyses, and created the graphical output. Both authors contributed to the writing of the manuscript

*Acknowledgements.*  This work was supported by the European Research Council (COEVOLVE, 313797) and the Swiss National Science Foundation (ACACIA, CR10I2_146314). We thank Shawn Koppenhoefer for assistance compiling and querying the weather databases and Alexis Berne and Grégoire Mariéthoz for helpful suggestions on the analyses. We are grateful to NOAA NCDC and the University of Washington for providing free of charge the GHCN-Daily and EECRA databases, respectively.

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
