# Peer review of "A globally calibrated scheme for generating daily meteorology from monthly statistics: Global-WGEN (GWGEN) v1.0"

_Geoscientific Model Development, 2017_

## Referee Comment (RC1) · Anonymous Referee #1 · 2 May 2017

**1  A globally calibrated scheme for generating daily meteorology from monthy statistics:Global-WGEN (GWGEN) v1.0**

This paper deals with the problem of constructing a statistical weather generator for obtaining / generating daily meteorology based on monthly statistics of the following five variables, precipitation, min temperature, max temperature, cloudness and wind speed. The data used are daily entries but the model is built so that the model parameters at the daily resolution to be expressed and therefore possible to compute from monthly statistics.

GWGEN is based on the WGEN by [7] , but diverges from it by using a 2nd order Markov chain and a hybrid model with Gaussian and GP distributions for modelling the precipitation occurrence and amount respectively. The latter model has been in [4] and [6]. Here though references are missing since both these models have been studied and applied extensively to different data sets.

The problem this work deals with is an interesting one. Unfortunatelly the presentation is lacking in clarity and probabilistic rigor. Moreover the biggest limitation of the weather generator, which is also mentioned by the authors, is the absence of the spatial factor. The spatial aspect of the problem is essential since there is spatial dependence between each variable but also between the different variables and the weather at one location today moves to another nearby location tomorrow.

Some specific comments follow:

- Section 2.2.1. The first line is rather strange if one takes into account that this is actually the frequential definition of probability. The probability that a given day is wet is defined as
$$P(\text{wet}) = \frac{\text{\# wet days}}{\text{\# total days}},$$
  for a specific month and station, so obviously there is a strong relationship. They represent exactly the same thing.

- Use of a 2nd order M-C gives better fit and results when modelling precipitation, see for example the study done in [5]. A 2nd order M-C is characterised by the transition probabilities $p_{ijk}, i, j, k = 0, 1$, therefore a total of 8 transition probabilities. I understand that the authors are interested only at the event of a wet day, i.e. we need $p_{ij1}, i, j = 0, 1$. These probabilities would be : $p_{001}, p_{101}, p_{011}, p_{111}$. Could the authors explain why instead of the last two they model $p_{11}$?

- In Fig.2 the fit is per station in a given month or for all months and stations together? The authors could do a better job explaining waht it is plotted in every Figure.

- In section 2.2.2, line 11: The strong relation... The mean of a gamma random variable equals the product of its two parameters. I.e $E(\Gamma) = \alpha\theta$.

- Cleary the extreme values cannot be modelled using the Gamma distribution, it is not suitable for this.

- Page 8, line 9 : I would prefer the use of the term density since distribution is usually reserved for the cumulative distribution function.

- How was the estimation of the Gamma distribution parameters performed? If the authors used likelihood, how did they deal with the fact that the the excesses above level $\mu$ are modelled as GP. [2] and [1] suggest a type of modeified likelihood that treats the excesses as sensored data.

- If $\alpha$ and $\theta$ are estimated by fitting the distribution to the data, then $E(\Gamma) = \bar{p} = \alpha\theta$. So what exactly is modelled by (7) and (8)? I am confused about what the authors are trying to achieve here.

- Fig. 11 Right. The data do not seem to be in a linear relation here. I think the authors shoud try some other relation or transformation also.

- Section 2.2.6. I could appreciate some comments on why the matrices $A$ and $B$ are needed and what they actually represent or try to model. Moreover, are the matrices estimated for each station and for every month? I assume that they are estimated using all months and stations together? How is something like this justified?

- What is the exact length of each simulation record? When we compare simulated versus observed records I assume that the simulated records are of exactly the same length as he observed ones?

- The authors notice that the gamma distribution does not perform so well for low values. Maybe it would make sense to stich another distribution for these low values, in the same fashion they did for the high ones.

- Section 2.5 The choice of the threshold $\mu$ is a rather difficult one, see for example [3]. The problem is that the fitting of the GP by likelihood, is based on the assumption that the excesses above level $\mu$ are independent and identically distributed. A rather difficult to satisfy assumption. If the level $\mu$ is chosen too low this will result to too many excesses that will be probably dependent. If it is chosen too high that would result to too few excesses to make any kind of reasonable fitting. Moreover, I think the use of a global threshold is oversimplifying.

**References**

[1] A. Baxevani and J. Lennartsson. A spatiotemporal precipitation generator based on a censored latent Gaussian field. *Water Resources Reseach*, 51:4338–4358, 2015.

[2] M. Durban and C. A. Glasbey. Weather modelling using a multivariate latent Gaussian model. *Agricultural and Forest Meteorology*, 109:187–201, 2001.

[3] A. Frigessi, O. Haug, and H. Rue. A dynamic mixture model for unsupervised tail estimation without threshold selection. *Extremes*, 5(3):219–235, 2002.

[4] M. Furrer, E. and R. Katz. Improving the simulation of extreme precipitation events by stochastic weather generators. *Water resources research*, 44(12):doi:10.1029/2008WR007316, 2008.

[5] J. Lennartsson, A. Baxevani, and D. Chen. Modelling precipitation in Sweden using multiple step Markov chains and a composite model. *Journal of Hydrology*, 363(1–4):42–59, 2008.

[6] N.M Neykov, P. N Neytchev, and W. Zucchini. Stochastic daily precipitation model with a heavy-tailed component. *Natural Haxards and Earth System Science*, 14:2321–2335, 2014.

[7] C. W Richardson. Stochastic simulation of daily precipitation, temperature, and solar radiation. *Water Resources Reseach*, 17:182–190, 1981.

---

## Referee Comment (RC2) · Anonymous Referee #2 · 12 May 2017

**Review of the paper "A globally calibrated scheme for generating daily meteorology from monthly statistics: Global-WGEN (GWGEN)"**

The paper presents a method to generate climate variables (rain, min and max temperatures, could cover and wind speed) from monthly to daily resolution. The scheme relies on the WGEN generator, and includes adaptations proposed by other authors (second-order Markov chain and hybrid Gamma-GP distribution for the precipitation, quantile-based bias correction for wind and min temperature). The objective of the paper is interesting and the method is useful, however the presentation has to be improved. It is hard for the reader to follow the multiple steps of the scheme. Moreover, some mistakes or bad choices for notations make the paper difficult to read, and some justifications/explanations are laking. Finally, I am not convinced that the extension to spatial autocorrelation would be so easy as described by the authors.

Some questions and remarks :

- Section 2.2.1 : Why are you not interested in $p_{011}$ and $p_{111}$?

- Figure 2 : There is no histogram.

- Table 1 and Figure 11 : you should mention the fact that $R^2$ are artificially high for models without constant because the $R^2$ formulae is modified for such models.

- In equation (5), can $\xi$ be equal to 0?

- Line 19 : you should explain how you estimate the parameters.

- Lines 20 to 25 : it is not clear, you should explain quickly what is done in Geng et al. (1986).

- Equation (10) : $x_{wet}$ and $x_{dry}$ have to be replaced by $\bar{x}_{wet}$ and $\bar{x}_{dry}$? Same remark for equation (11).

- Figure 5 and 7 : As you write, the adjustement is very bad. I think you should propose another way of fixing the standard deviations. You write "we believe that the error introduced by the poor linear fit is negligible", but this is not convincing.

- Equation (12) : please explain how this formulae has been chosen.

- Page 13 line 5 : $c_{sd,dry}$ has to be replaced by $\sigma_{c,dry}$, same for "wet".

- Equation (12) : $c$ has to be replaced by $c_{wet}$ or $c_{dry}$. Moreover, bars have to be added, since you describe mean cloud cover.

- Equation (13) : same remark, bars have to be added.

- Section 2.2.6 : Please describe how you add the residual noise in practice. It is described at the end of Algorithm 1 but it is not clear : residuals for one day are really computed from the residuals of the previous day as written line 18? If so, you should explain why.

- Section 2.5 : Can you present/discuss some references about the estimation problem of GP parameters?

- Section 3 : I think such a global presentation of the model should be given at the beginning of the paper in order to help the reader following al steps. Maybe with a schematic description?

- I think Sections 4 et 5 could be merged.

- Section 5 : I am not convinced that the introduction of a spatial autocorrelation field on the sequence of random numbers would solve the problem so easily. The spatial correlation will not be the same for the whole globe and for all variables, and may be hard to fix.

---

## Editor Comment (EC1) · C. van Heerwaarden (Editor) · 13 May 2017

Based on the two reviews, I conclude that there is a lack of clarity in the presentation. In addressing your replies, I suggest that you explain in detail to the points of the reviewers and also suggest how you will improve the paper. The reviews provide good guidance for improvement.

One aspect that needs to be addressed is the horizontal component. I agree with the reviewers that a substantial amount of work needs to be devoted to address this comment. Ideally, you implement the spatial autocorrelation and add it to the paper, as you claim that this is feasible.

---

## Author Comment (AC1) · 30 Jun 2017

We thank the anonymous reviewer for his helpful comments to our manuscript. The manuscript for GWGEN, a weather generator for precipitation, temperature, cloud fraction and wind speed using a hybrid Gamma-GP distribution, a hybrid-order Markov Chain and a cross correlation approach) has been revised and improved.

In summary, a bug has been fixed that now makes the quantile-based bias correction for the minimum temperature redundant and instead another quantile-based bias correction for the wind speeds intercept has been implemented to further improve the results. Furthermore we made several attempts to improve the manuscript text for clar-

ity and style. This includes a schematic representation of the workflow, changes in the structure of the paper, more explanations to the figures and a fix of the notation in the equations.

The spatial autocorrelation, however, that has also been addressed by the other reviewer and the editor is, to our believe, beyond the scope of this manuscript. Although we think that it is possible, we agree with the reviewers that it is not that simple and subject to further research. Already for the technical aspect we would need a few months to fix this issue. Nevertheless we think that this does not affect the utility of the weather generator for a wide range of applications.

Detailed responses to the comments of the reviewer can be found below.

**Responses**

**Reviewer** Section 2.2.1. The first line is rather strange if one takes into account that this is actually the frequential definition of probability. The probability that a given day is wet is defined as

$$P(\text{wet}) = \frac{\#\text{wet days}}{\#\text{total days}}, \tag{1}$$

for a specific month and station, so obviously there is a strong relationship. They represent exactly the same thing.

**Response** We edited the text to acknowledge this fact.

**Reviewer** Use of a 2nd order MC gives better fit and results when modelling precipitation, see for example the study done in Lennartsson et al. (2008). A 2nd order MC is characterised by the transition probabilities $p_{ijk}; i, j, k = 0, 1$, therefore a total of 8 transition probabilities. I understand that the authors are interested only at the event of a wet day, i.e. we need $p_{ij1}; i; j = 0, 1$. These probabilities would

be : $p_{001}, p_{101}, p_{011}, p_{111}$. Could the authors explain why instead of the last two they model $p_{11}$?

**Response** Any model development requires choices and trade-offs between absolute realism and computational demand. Following the analyses and recommendation of Wilks (1999), we use a hybrid-order model that retains first-order Markov dependence for wet spells but allows higher-order dependence for dry sequences as a compromise between effectiveness and simplicity. This approach therefore only uses the probabilities up to the last wet day, which are $p_{11}$ and $p_{101}$, as well as $p_{001}$ for a dry sequence. Using this, the MC only needs 3 probabilities instead of 4. We will include this explanation in the paper.

**Reviewer** In Fig.2 the fit is per station in a given month or for all months and stations together? The authors could do a better job explaining waht it is plotted in every Figure.

**Response** We clarify our methodology by providing the following description in the text

*We perform this analysis on a station and month-wise basis, i.e., we first extract each of the (complete) Januaries, Februaries, etc. for a given station, and then merge all of the Januaries (Februaries, Marches, etc...) for this station into a single series representing each month. [. . . ] Merging months over several years is particularly important for stations that have relatively little precipitation in a given month; for example, it could take several years of observations to observe a single ($p_{101}$) event. The final transition probabilities were then regressed against the fraction of days in the month with precipitation, which show the characteristic linear relationship described by Geng et al. (1986)*

**Furthermore**, the figure captions now include a new short description for clarification:

*The underlying data for the fits correspond to the means of the the multi-year series for each month for each station.*

**Reviewer** In section 2.2.2, line 11: The strong relation... The mean of a gamma random variable equals the product of its two parameters. i.e $E(\Gamma) = \alpha\theta$

**Response** We include following explanation in the revised manuscript:

*The strong relationship between the gamma scale parameter and the mean precipitation on wet days noted by Geng et al. (1986) makes generation of precipitation amounts with only monthly input data feasible. It is based upon the fact that the expected value of a gamma random variable equals the product of its two parameters. i.e $E(\Gamma) = \alpha\theta$.*

**Reviewer** Cleary the extreme values cannot be modelled using the Gamma distribution, it is not suitable for this.

**Response** We agree, and as noted, we adopt the hybrid Gamma-GP approach to capture high precipitation amounts as suggested by several previous studies.

**Reviewer** Page 8, line 9 : I would prefer the use of the term density since distribution is usually reserved for the cumulative distribution function.

**Response** It has been changed to probability density function (pdf)

**Reviewer** How was the estimation of the Gamma distribution parameters performed? If the authors used likelihood, how did they deal with the fact that the the excesses above level are modelled as GP. Durban and Glasbey (2001) and Baxevani and Lennartsson (2015) suggest a type of modeified likelihood that treats the excesses as sensored data.

**Response** We used all of the data in fitting the Gamma distribution using likelihood. We acknowledge that there could have been different approaches to this problem, including censuring data above the threshold, but the final results of our model as presented are acceptable to us. We clarify this point in the text when discussing the fitting procedure.

**Reviewer** If $\alpha$ and $\theta$ are estimated by fitting the distribution to the data, then $E(\Gamma) = \bar{p} = \alpha\theta$. So what exactly is modelled by (7) and (8)? I am confused about what the authors are trying to achieve here.

**Response** We explain that the resulting $\alpha$ in our calculations ends up being a constant, effectively the slope of the relationship between the Gamma scale parameter and $\bar{p}$, and revise equation (8) to reflect this fact.

**Reviewer** Fig. 11 Right. The data do not seem to be in a linear relation here. I think the authors shoud try some other relation or transformation also.

**Response** We agree and use a third order polynomial now which significantly improves the relationship

**Reviewer** Section 2.2.6. I could appreciate some comments on why the matrices A and B are needed and what they actually represent or try to model. Moreover, are the matrices estimated for each station and for every month? I assume that they are estimated using all months and stations together? How is something like this justified?

**Response** We added additional clarification and explanation on this point at the beginning of the relevant section.

**Reviewer** What is the exact length of each simulation record? When we compare simulated versus observed records I assume that the simulated records are of exactly the same length as he observed ones?

**Response** While the lengths of the observed meteorological records differ for each station, in our model evaluation, we simulate a daily weather record that is exactly as long as the input monthly weather observations. We clarify this point in the text.

**Reviewer** The authors notice that the gamma distribution does not perform so well for low values. Maybe it would make sense to stich another distribution for these low values, in the same fashion they did for the high ones.

**Response** Ecological and hydrological significance of very low precipitation is small, also we are close to the precision of the measurements. For the sake of model parsimony, we use the current methodology, as also suggested by several other authors.

**Reviewer** Section 2.5 The choice of the threshold $\mu$ is a rather difficult one, see for example Frigessi et al. (2002). The problem is that the fitting of the GP by likelihood, is based on the assumption that the excesses above level $\mu$ are independent and identically distributed. A rather difficult to satisfy assumption. If the level $\mu$ is chosen too low this will result to too many excesses that will be probably dependent. If it is chosen too high that would result to too few excesses to make any kind of reasonable fitting. Moreover, I think the use of a global threshold is oversimplifying.

**Response** We agree to this point. However, although we did fit the GP to our parameterization data above the threshold, this information could not be used. Instead, we decided to use constant parameters for the GP shape and the threshold and make a sensitivity analysis (previous section 2.5). The reason for this is, that after extensive data analysis, we could not find any good relationship between $\xi, \mu$ and the input data for our weather generator. In fact, as stated in the text, we also tried a varying threshold such that the GP distribution is used, when the Gamma random variable exceeds a certain percentile, but we could not find any improvement.

Therefore we could not justify a varying $\xi$ and $\mu$ although we acknowledge the fact, that this is oversimplifying and we clarified this in the discussion. At the moment this is the best we can do and the results are nevertheless better compared

to using the Gamma distribution alone and, indeed, they are surprisingly good.

**References**

Baxevani, A. and Lennartsson, J.: A spatiotemporal precipitation generator based on a censored latent Gaussian field, Water Resources Research, 51, 4338–4358, doi:10.1002/2014WR016455, http://dx.doi.org/10.1002/2014WR016455, 2015.

Durban, M. and Glasbey, C.: Weather modelling using a multivariate latent Gaussian model, Agricultural and Forest Meteorology, 109, 187 – 201, doi:http://dx.doi.org/10.1016/S0168-1923(01)00268-4, http://www.sciencedirect.com/science/article/pii/S0168192301002684, 2001.

Frigessi, A., Haug, O., and Rue, H.: A Dynamic Mixture Model for Unsupervised Tail Estimation without Threshold Selection, Extremes, 5, 219–235, doi:10.1023/A:1024072610684, http://dx.doi.org/10.1023/A:1024072610684, 2002.

Geng, S., Devries, F. W. T. P., and Supit, I.: A Simple Method for Generating Daily Rainfall Data, Agricultural and Forest Meteorology, 36, 363–376, doi:10.1016/0168-1923(86)90014-6, <GotoISI>://WOS:A1986C086500007, 1986.

Lennartsson, J., Baxevani, A., and Chen, D.: Modelling precipitation in Sweden using multiple step markov chains and a composite model, Journal of Hydrology, 363, 42 – 59, doi:10.1016/j.jhydrol.2008.10.003, http://www.sciencedirect.com/science/article/pii/S0022169408004848, 2008.

Wilks, D. S.: Interannual variability and extreme-value characteristics of several stochastic daily precipitation models, Agricultural and Forest Meteorology, 93, 153–169, doi:10.1016/S0168-1923(98)00125-7, <GotoISI>://WOS:000079269800001, 1999.

---

## Author Comment (AC2) · 30 Jun 2017

We thank the anonymous reviewer for his helpful comments to our manuscript. The manuscript for GWGEN, a weather generator for precipitation, temperature, cloud fraction and wind speed using a hybrid Gamma-GP distribution, a hybrid-order Markov Chain and a cross correlation approach) has been revised and improved.

In summary, a bug has been fixed that now makes the quantile-based bias correction for the minimum temperature redundant and instead another quantile-based bias correction for the wind speeds intercept has been implemented to further improve the results. Furthermore we made several attempts to improve the reading. This includes

a schematic representation of the workflow, changes in the structure of the paper, more explanations to the figures and a fix of the notation in the equations.

Detailed responses to the comments of the reviewer can be found below.

**Responses**

**Reviewer** Section 2.2.1 : Why are you not interested in p011 and p111?

**Respones** We use the hybrid-order Markov Chain as recommended by Wilks (1999) as a tradeoff between accuracy and simplicity. This model retains first-order Markov dependence for wet spells but allows second-order dependence for dry sequences. It therefore only requires the three parameters $p_{11}, p_{001}$ and $p_{101}$, i.e. the probabilities up to the last wet day. We specified it explicitly now.

**Reviewer** Figure 2 : There is no histogram.

**Respones** It is a 2D histogram, i.e. the value for each grid cell represents the sum of observations that fall into this cell. To clarify this point, we replaced the word histogram with *density plot*.

**Reviewer** Table 1 and Figure 11 : you should mention the fact that R2 are artificially high for models without constant because the R2 formulae is modified for such models.

**Response** We now acknowledge this fact in the paper.

**Reviewer** In equation (5), can $\xi$ be equal to 0?

**Response** Yes, in this case, $g(x) = \frac{1}{\sigma}e^{-\frac{x-\mu}{\sigma}}$. We added it to the equation.

**Reviewer** Line 19 : you should explain how you estimate the parameters.

**Response** The parameters of the Gamma distribution are estimated using likelihood. We included it in the text.

**Reviewer** Lines 20 to 25 : it is not clear, you should explain quickly what is done in Geng et al. (1986)

**Response** We clarified in the text that the expected value of the Gamma distribution is the product of the shape and scale parameter, i.e. $E(\Gamma) = \alpha\theta$. This justifies equation (7) and (8).

**Reviewer** Equation (10) : $x_{wet}$ and $x_{dry}$ have to be replaced by $\bar{x}_{wet}$ and $\bar{x}_{dry}$? Same remark for equation (11).

**Response** They have been replaced.

**Reviewer** Figure 5 and 7 : As you write, the adjustement is very bad. I think you should propose another way of fixing the standard deviations. You write "we believe that the error introduced by the poor linear fit is negligible", but this is not convincing.

**Response** We agree with this comment and in our revised version of the model have completely re-thought the way we estimate the SD of temperature. These changes are described in the revised manuscript. In summary, instead of a linear fit, we now use a combination out two polynomials combined with a linear extrapolation at the cold and warm extremes. For minimum temperature that means, that values below $-40°C$ and above $25°C$ are linearly extrapolated, whereas $\sigma_{T_{\min},\text{dry/wet}}$ between $-40°C < \bar{T}_{\min,\text{dry/wet}} \leq 0°C$ and $0°C < \bar{T}_{\min,\text{dry/wet}} \leq 25°C$ is modeled by two different polynomials of order 5. We use the same methodology for maximum temperature with $-30°C$ instead of $-40°C$, and $35°C$ instead of $25°C$ (see attached figures below).

Although this procedure is more complicated, results in a significant improvement in the simulation of extreme temperatures, and an overall improvement in the simulation of daily temperature.

**Reviewer** Equation (12) : please explain how this formulae has been chosen.

**Response** We wrote in the original manuscript that we chose the shapes of the curves to reflect the phenomenon that wet days should be cloudier on average than dry days. We clarify our selection of these equations based on the constraints presented by the variable, e.g., that cloud fraction on wet days must be 0 when total mean cloud fraction is 0 and 1 when the total mean cloud fraction is 1. We used a qualitative graphical analysis to develop "best guess" equations that had the desired shape.

**Reviewer** Page 13 line 5 : $c_{sd,dry}$ has to be replaced by $\sigma_{c,dry}$, same for "wet".

**Response** They have been replaced

**Reviewer** Equation (12) : $c$ has to be replaced by $c_{wet}$ or $c_{dry}$. Moreover, bars have to be added, since you describe mean cloud cover.

**Response** Bars have been added but $c$ (or rather $\bar{c}$) should not be replaced by $\bar{c}_{wet}$ or $\bar{c}_{dry}$ since in this case we use the monthly mean cloud fraction $\bar{c}$ to calculate the mean of the wet ($\bar{c}_{wet}$) or dry ($\bar{c}_{dry}$) days in the month

**Reviewer** Equation (13) : same remark, bars have to be added.

**Response** They have been added

**Reviewer** Section 2.2.6 : Please describe how you add the residual noise in practice. It is described at the end of Algorithm 1 but it is not clear : residuals for one day are really computed from the residuals of the previous day as written line 18? If so, you should explain why.

**Response** Yes, they are computed from the previous day. We extended the explanation in the corresponding section. The procedure is based upon Matalas (1967) and

preserves the serial and the cross correlation between the simulated variables. Otherwise longer periods of, e.g. warm temperatures, could not be simulated.

**Reviewer** Section 2.5 : Can you present/discuss some references about the estimation problem of GP parameters?

**Response** We acknowledge that choosing globally fixed parameters for the location parameter $\mu$ and the threshold $\xi$ is a simplified aspect of our model (we also state that in the revised manuscript) and is generally not easy (e.g. Davison and Smith, 1990; Neykov et al., 2014; Rootzén and Tajvidi, 1997). Frigessi et al. (2002) suggest to use a dynamical mixture model instead of a fixed threshold. Rust et al. (2009) vary the parameters with seasonality.

However, to our knowledge, no global application of these methods has been published and for now, therefore we stick to the simplest methodology with fixed parameters that are based on a sensitivity analysis (described in detail in section 3.5 of the revised manuscript). We also performed extensive data analysis in an attempt to correlate the GP parameters with other input variables for our weather generator, but could not find any relationship that would allow us to perform a dynamic calculation. As we say in the discussion section, this is subject to further improvement, but, nevertheless, despite the simplicity of our parameterization of the hybrid Gamma-GP distribution, the results are excellent.

**Reviewer** Section 3 : I think such a global presentation of the model should be given at the beginning of the paper in order to help the reader following al steps. Maybe with a schematic description?

**Response** It has been moved and a schematic of the workflow has been added

**Reviewer** I think Sections 4 et 5 could be merged.

**Response** They have been merged.

**Reviewer** Section 5 : I am not convinced that the introduction of a spatial autocorrelation field on the sequence of random numbers would solve the problem so easily. The spatial correlation will not be the same for the whole globe and for all variables, and may be hard to fix.

**Response** We agree, but also continue to believe that implementation of spatial autocorrelation is beyond the scope of the current manuscript, and does not affect the utility of the weather generator for a wide range of applications. We clarify the challenge of implementing autocorrelation in the manuscript, and remove our specific recommendation, tending to agree with the reviewer that, although possible, the solution would not be **that** simple.

**References**

Davison, A. C. and Smith, R. L.: Models for Exceedances over High Thresholds, Journal of the Royal Statistical Society. Series B (Methodological), 52, 393–442, http://www.jstor.org/stable/2345667, 1990.

Frigessi, A., Haug, O., and Rue, H.: A Dynamic Mixture Model for Unsupervised Tail Estimation without Threshold Selection, Extremes, 5, 219–235, doi:10.1023/A:1024072610684, http://dx.doi.org/10.1023/A:1024072610684, 2002.

Geng, S., Devries, F. W. T. P., and Supit, I.: A Simple Method for Generating Daily Rainfall Data, Agricultural and Forest Meteorology, 36, 363–376, doi:10.1016/0168-1923(86)90014-6, <GotoISI>://WOS:A1986C086500007, 1986.

Matalas, N. C.: Mathematical assessment of synthetic hydrology, Water Resources Research, 3, 937–945, doi:10.1029/WR003i004p00937, http://dx.doi.org/10.1029/WR003i004p00937, 1967.

Neykov, N. M., Neytchev, P. N., and Zucchini, W.: Stochastic daily precipitation model with a heavy-tailed component, Natural Hazards and Earth System Science, 14, 2321–2335, doi:10.5194/nhess-14-2321-2014, 2014.

Rootzén, H. and Tajvidi, N.: Extreme value statistics and wind storm losses: A case study,
Scandinavian Actuarial Journal, 1997, 70–94, doi:10.1080/03461238.1997.10413979, http://dx.doi.org/10.1080/03461238.1997.10413979, 1997.

Rust, H. W., Maraun, D., and Osborn, T. J.: Modelling seasonality in extreme precipitation, The European Physical Journal Special Topics, 174, 99–111, doi:10.1140/epjst/e2009-01093-7, http://dx.doi.org/10.1140/epjst/e2009-01093-7, 2009.

Wilks, D. S.: Interannual variability and extreme-value characteristics of several stochastic daily precipitation models, Agricultural and Forest Meteorology, 93, 153–169, doi:10.1016/S0168-1923(98)00125-7, <GotoISI>://WOS:000079269800001, 1999.
* * *
[Figure]

**Fig. 1.** Correlation of standard deviation of min. temperature to the monthly mean

[Figure]

**Fig. 2.** Correlation of standard deviation of max. temperature to the monthly mean

---

## Author Response (AR2)

Dear Dr van Heerwaarden,

Thank you for taking the time to evaluate our manuscript "A globally calibrated scheme for generating daily meteorology from monthly statistics: Global-WGEN (GWGEN) v1.0" (gmd-2017-42). We greatly appreciate your comments and those from the reviewers, all of which have substantially improved our manuscript.

Now our manuscript has been through two review cycles. The brief and unconstructive comments of Referee #1 on our revised manuscript notwithstanding, you agree that we made a satisfactory effort to respond to the reviewers' comments and edit our manuscript following the first round of reviews, and we thank you for that.

However, you now asking us to add major new functionality to our model, i.e., simulating spatial autocorrelation, as a requirement for publishing the manuscript in GMD. In this, we must respectfully disagree with your decision. At several points in the manuscript, including in the abstract and in an extensive discussion in Section 4, we very clearly state that our model does not include spatial autocorrelation among its features. We provide examples of the type of applications for which this might make the model unsuitable, but point out that in several other applications spatial autocorrelation in daily meteorological forcing is unnecessary. For example, in most dynamic global vegetation models there is no lateral connection between gridcells, and therefore an explicit representation of spatial autocorrelation in the driving daily meteorological data would have no effect on the model output. As another example, one of the primary uses of weather generators in the past has been to drive crop models for a particular location, with a single point of meteorological forcing. For this, our GWGEN model in its current form would be entirely suitable, and a substantial improvement over many of the weather generators that are in use today, because of our global parameterization, simulation of new weather variables, and extensive and thorough evaluation. We further point out that if the monthly data used to drive the model are spatially autocorrelated – this would be the case when using gridded climatology for example – then the result of the weather generator will also preserve this autocorrelation, at least when the model results analyzed on monthly or longer timescales.

We agree that spatial autocorrelation could be a valuable feature for a future version of the GWGEN model, and we write this in our manuscript. However, description, implementation, and evaluation of such a feature would go well beyond the current manuscript, make the paper much longer and more difficult to read, and as noted above, be unnecessary for many current applications of weather generators. We therefore do not believe that implementation of this new feature should be a requirement for publishing our manuscript in GMD. We have made every effort to carefully follow all of the guidelines for publishing a model description paper as described in the GMD information for authors.

We do not feel that we are hiding anything in our model description, or misrepresenting the model's use case. We are, however, at your request, willing to further revise our manuscript text to be even more explicit about the lack of the spatial autocorrelation feature, and therefore provide a strong caveat to potential users about its usefulness for certain specific applications.

We would be pleased to discuss this issue with you further, and incorporate any other specific constructive feedback you may have.

Thank you for your consideration.

Sincerely,

Philipp Sommer and Jed Kaplan

[revised manuscript text omitted]
_{\mathrm{in}}\,[\mathrm{mm}]$, cloud cover fraction $c_{\mathrm{in}}$, minimum ($T_{\mathrm{min,in}}\,[^\circ\mathrm{C}]$) and maximum ($T_{\mathrm{max,in}}\,[^\circ\mathrm{C}]$) temperature, wind speed $w_{\mathrm{in}}\,[\mathrm{m/s}]$, number of wet days $n_{\mathrm{in}}$

**Output:** daily $P_i\,[\mathrm{mm/d}], c_i, T_i\,[^\circ\mathrm{C}], w_i\,[\mathrm{m/s}]$ and the wet/dry state $s_i \in \{0,1\}$

1: **for** month $m$ in $input$ **do**

2:    smooth the monthly data using Rymes and Myers (2001)

3:    Set $j = 0, \chi = 0$

4:    **while** $j \equiv 0$ or $\left|\sum_{d_i \in m} P_i - P_{\mathrm{in}}\right| > \min\left(5\% \cdot P_{\mathrm{in}}, 0.5mm\right)$ or $|n_{\mathrm{sim}} - n_{\mathrm{in}}| > 1$ **do**

5:       **for** day $d_i$ in $m$ **do**

6:          Calculate $p_{11}, p_{101}, p_{001}$ after equations (1) - (3) using $n$ {Precipitation occurence after Wilks (1999a)}

7:          Use the Markov chain to determine whether $d_i$ is wet ($s_i = 1$) or dry ($s_i = 0$)

8:          **if** $s_i = 1$ **then**

9:             Calculate $\theta, \alpha$ and $\sigma$ via eq. (7)-(9) {Precipitation amount after Neykov et al. (2014)}

10:             Draw a random number $P_i$ from the Gamma-GP distribution, eq. (6)

11:             Set $T_{\mathrm{min},i} = T_{\mathrm{min,wet}}, T_{\mathrm{max},i} = T_{\mathrm{max,wet}}, c_i = c_{\mathrm{wet}}, w_i = w_{\mathrm{wet}}$ from eq. (10) and (12) and tables 1, 3

12:             Set $\sigma_{T_{\mathrm{min}},i} = \sigma_{T_{\mathrm{min,wet}}}, \sigma_{T_{\mathrm{max}},i} = \sigma_{T_{\mathrm{max,wet}}}, \sigma_{w,i} = \sigma_{w,\mathrm{wet}}, \sigma_{c,i} = \sigma_{c,\mathrm{wet}}$ from eq. (11), (13) and (14) and tables 1, 2, 3

13:          **else**

14:             Set $P_i = 0\,\mathrm{mm/d}$

15:             Set $T_{\mathrm{min},i} = T_{\mathrm{min,dry}}, T_{\mathrm{max},i} = T_{\mathrm{max,dry}}, c_i = c_{\mathrm{dry}}, w_i = w_{\mathrm{dry}}$ from eq. (10) and (12) and tables 1, 3

16:             Set $\sigma_{T_{\mathrm{min}},i} = \sigma_{T_{\mathrm{min,dry}}}, \sigma_{T_{\mathrm{max}},i} = \sigma_{T_{\mathrm{max,dry}}}, \sigma_{w,i} = \sigma_{w,\mathrm{dry}}, \sigma_{c,i} = \sigma_{c,\mathrm{
[revised manuscript text omitted]

---

## Author Response (AR3)

**Response to revision 3: A globally calibrated scheme for generating daily meteorology from monthly statistics: Global-WGEN (GWGEN) v1.0**

Dear Dr van Heerwaarden,

Thank you again for your comments and reconsideration of our manuscript. Following your recommendations, we have now added a new section to the paper (section 4, "Limitations") and are more explicit both in that section and in the discussion about the applications for which our model may or may not be suitable.

We hope this satisfies your requests and are happy to answer any further questions you may have.

Sincerely,

Philipp Sommer and Jed Kaplan

[revised manuscript text omitted]

---

## Author Response (AR4)

**Response to acceptance of final publication: A globally calibrated scheme for generating daily meteorology from monthly statistics: Global-WGEN (GWGEN) v1.0**

Dear Dr van Heerwaarden,

Thank you for accepting our publication and again, for your comments and your help. For the last version, we made some minor typo corrections and provided a DOI for the codebase (https://doi.org/10.5281/zenodo.889213) which is also listed in the Code Availability section at the end of the paper.

Sincerely,

Philipp Sommer and Jed Kaplan

[revised manuscript text omitted]